# DLM-Scope: Mechanistic Interpretability of Diffusion Language Models via Sparse Autoencoders

**Xu Wang** [1 2]  **Bingqing Jiang** [1]  **Yu Wan** [2]  **Baosong Yang** [2]  **Lingpeng Kong** [1]  **Difan Zou** [1]

⬡ https://github.com/Xu0615/SAE4DLM

🤗 https://huggingface.co/DLM-Scope

## Abstract

Sparse autoencoders (SAEs) have become a standard tool for mechanistic interpretability in autoregressive large language models (LLMs), enabling researchers to extract sparse, human-interpretable features and intervene on model behavior. Recently, as diffusion language models (DLMs) have become an increasingly powerful and promising alternative to the autoregressive LLMs, it is essential to develop tailored mechanistic interpretability tools for this emerging class of models. In this work, we present **DLM-Scope**, the first SAE-based interpretability framework for DLMs, and demonstrate that trained Top-K SAEs can faithfully extract sparse, interpretable features. Notably, we find that inserting SAEs affects DLMs differently than autoregressive LLMs: while SAE insertion in LLMs typically incurs a loss penalty, in DLMs it can reduce cross-entropy loss when applied to early layers, a phenomenon absent or markedly weaker in LLMs. Additionally, SAE features in DLMs enable more effective diffusion-time interventions, often outperforming LLM steering. Moreover, we pioneer certain new SAE-based research directions for DLMs: we show that SAEs can provide useful signals for DLM decoding order; and the SAE features are stable during the post-training phase of DLMs. Our work establishes a foundation for mechanistic interpretability in DLMs and shows a great potential of applying SAEs to DLM-related tasks and algorithms.

[1]The University of Hong Kong [2]Tongyi Lab, Alibaba Group Inc. Correspondence to: Xu Wang <sunny615@connect.hku.hk>, Difan Zou <dzou@hku.hk>.

*Proceedings of the 43rd International Conference on Machine Learning*, Seoul, South Korea. PMLR 306, 2026. Copyright 2026 by the author(s).

## 1. Introduction

Sparse autoencoders (SAEs) have become a widely used tool for mechanistic interpretability in autoregressive large language models (LLMs) (Karvonen et al., 2024; Marks et al., 2024), extracting sparse, human-meaningful features that help reveal internal representations (Templeton et al., 2024; Cunningham et al., 2023; Bricken et al., 2023). Beyond analysis, SAE features have been used in applications such as reducing hallucinations (Ferrando et al., 2025) and mitigating biases (Durmus et al., 2024).

Meanwhile, diffusion language models (DLMs) are becoming increasingly competitive for text understanding and generation (Gong et al., 2023; Wu et al., 2023; Xu et al., 2025), with recent results suggesting favorable scaling behavior (Nie et al., 2025a; Bie et al., 2025). Moreover, diffusion is not inherently uninterpretable: in text-to-image models, SAEs expose causal concept factors and enable reliable concept editing (Surkov et al., 2025; He et al., 2025). In contrast, the interpretability of DLMs remains limited; therefore, it is imperative to develop tools that facilitate a deeper inspection and understanding of these modern models.

To this end, a straightforward idea is to extend the SAE-based interpretability interface that has worked well in practice for LLMs to DLMs. However, transferring the LLM-SAEs recipe to DLM-SAEs is not plug-and-play, due to the fundamental difference in their mechanism. In particular, DLMs require certain design choices that are absent in LLMs: (i) selecting which token positions provide training activations under diffusion corruption, and (ii) defining inference-time steering policies that act repeatedly across denoising steps. This requires delicate configurations on the design of SAE training and inference for DLMs.

In this work, we present **DLM-Scope**, the first SAE-based mechanistic interpretability interface for DLMs. We develop a comprehensive training and inference framework for Top-$K$ SAEs (Gao et al., 2024) on DLMs, including Dream-7B (Ye et al., 2025) and LLaDA-8B (Nie et al., 2025b). Furthermore, we verify the utility of these SAEs under sparsity constraints and leverage features for both diffusion-time

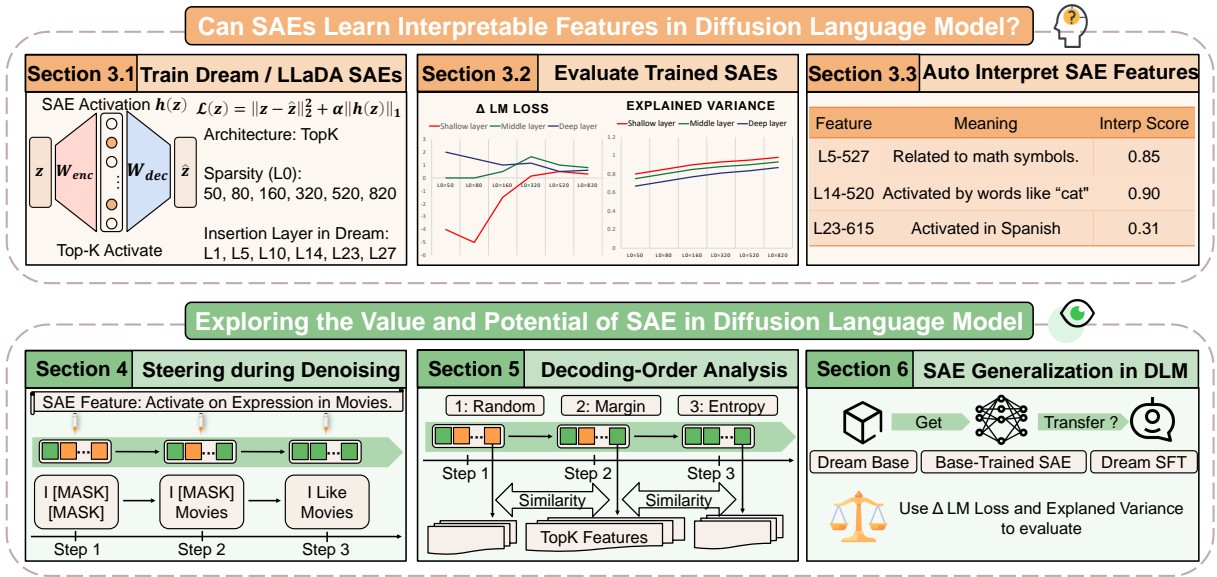

*Figure 1.* **DLMscope pipeline. Top (orange): DLM-SAE training and validation. Left:** Top-$K$ SAEs are trained on Dream/LLaDA.
**Middle:** They are evaluated via sparsity-fidelity. **Right:** They are auto-interpreted by generating explanations and interpretability scores.
**Bottom (green): The value of DLM-SAEs. Left:** Feature steering is applied across denoising steps. **Middle:** Different decoding-orders
are analyzed by tracking Top-$K$ feature dynamics. **Right:** Cross-training transfer is tested by applying base-trained SAEs to DLM-SFT.

steering and tracking representation dynamics across different remasking orders. Additionally, we evaluate transferability by applying base-trained SAEs to instruction-tuned DLM, showing the generality of the learned features during the post-training process of DLM. Our contributions are summarized as follows (also detailed in Fig. 1):

1. **SAEs trained on DLMs are usable for mechanistic analyses.**

   (§3.1) *We design the training strategy and objective for SAEs in DLMs.* We introduce DLM-SAE training by sampling activations from denoising states and explicitly choosing positions under corruption.

   (§3.2) *DLM-SAEs achieve a favorable sparsity-fidelity profile.* Notably, in early layers we observe regimes where inserting an SAE into DLMs can *reduce* masked-token cross-entropy loss.

   (§3.3) *SAEs Extract Interpretable Features in DLMs.* We apply automated feature interpretation (Paulo et al., 2025) to DLM-SAE latents, producing understandable explanations and interpretability scores.

2. **SAEs enable effective steering, remasking strategy analysis, and transfer to instruction-tuned DLM.**

   (§4) *We design DLM-specific per-step steering policies.* We find SAE features enable effective diffusion-time interventions that often outperform LLM steering.

   (§5) *SAEs provide an interpretable quantification of different remasking strategies.* Using our proposed metrics ($S_{\ell,k,i}^{\mathrm{pre}}(\mathcal{O})$ and $D_{\ell,i}^{\mathrm{post}}(\mathcal{O})$), we observe semantic changes across decoding orders during denoising.

(§6) *Base→SFT transfer of DLM-SAEs.* We test whether base-trained SAEs remain faithful on instruction-tuned DLM, showing generality of SAEs in DLM.

## 2. SAE Training and Steering in DLMs

### 2.1. Preliminary

**Sparse Autoencoders (SAEs)**   SAEs act as microscopes for the dense, superposition-laden hidden states of language models. Mathematically, let $x \in \mathbb{R}^d$ denote a residual-stream activation at a fixed layer and token position. The goal of an SAE is to decompose this dense vector into a linear combination of interpretable directions. The SAE first encodes $x$ into sparse features $h \in \mathbb{R}^k$ (typically $k > d$) and then decodes back to a reconstruction $\hat{x} \in \mathbb{R}^d$. In particular, the SAE model is trained by the following loss function:

$$
\begin{aligned}
\mathcal{L}_{\mathrm{SAE}} &= \|x - \hat{x}\|_2^2 + \lambda\|h\|_1, \\
h &= \mathrm{ReLU}(W_E x + b_E), \quad \hat{x} = W_D h + b_D.
\end{aligned}
\tag{1}
$$

Eq. (1) trades off reconstruction error and sparsity, where $\lambda$ controls the $\ell_1$ penalty on features.

SAE feature steering is a causal intervention technique that modifies the model's internal processing by artificially activating specific interpretable concepts. This is achieved by modifying the residual stream with a decoder atom $v_f$ for a chosen feature $f$. Given a steering strength $\alpha$ and a per-sample scale $m_f$, we intervene as

$$
x^{\mathrm{steer}} = x + \alpha\, m_f\, v_f.
\tag{2}
$$

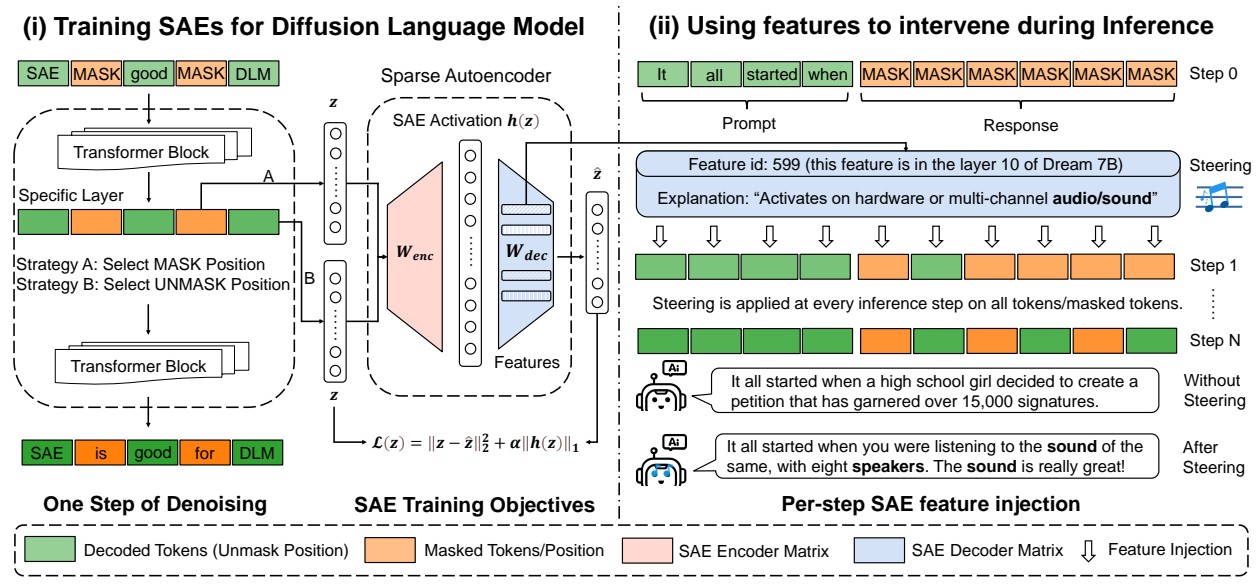

*Figure 2.* **DLM-SAE overview. Left: Training DLM-SAEs.** We collect residual-stream activations from one-step denoising inputs and train SAEs using two strategies: MASK-SAE or UNMASK-SAE. **Right: Diffusion-time feature steering.** During inference, we select feature $f$ and inject its decoder direction into the residual stream at every denoising step, either on all positions or update positions.

Eq. (2) pushes $x$ along the feature direction $v_f$ at a target layer and changes the output of the model.

**Diffusion Language Models (DLMs)** Unlike the autoregressive decoding in the standard LLMs, DLMs generate the tokens by iterative denoising and parallel decoding. Let $x_0 = (x_0^1, \ldots, x_0^N)$ be a length-$N$ token sequence sampled from the data distribution $q(x)$. DLM defines a noising process that produces a partially masked sequence $x_t \sim q(x_t \mid x_0)$ at mask rate $t \in (0, 1)$, and trains a mask predictor $p_\theta(\cdot \mid x_t)$ to recover the original tokens at masked positions:

$$\mathcal{L}_{\text{DLM}}(\theta) = \mathbb{E}_{x_0 \sim q(x),\, t \sim U(0,1),\, x_t \sim q(x_t|x_0)}$$
$$\left[ w(t) \sum_{i=1}^N \mathbf{1}[x_t^i = \texttt{[MASK]}] \cdot \underbrace{-\log p_\theta(x_0^i \mid x_t)}_{\text{cross-entropy}} \right] \quad (3)$$

Eq. (3) applies cross-entropy only to masked tokens, with timestep weight $w(t)$ (we use $w(t) = 1/t$).

During inference, DLMs denoise by repeatedly predicting tokens and remasking. At step $k$:

$$\tilde{\mathbf{x}}_0 \sim p_\theta(\cdot \mid \mathbf{x}_{t_k}),$$
$$\mathbf{x}_{t_{k-1}} = \text{Remask}(\mathbf{x}_{t_k}, \tilde{\mathbf{x}}_0; t_{k-1}). \quad (4)$$

Eq. (4) first samples a fully denoised prediction $\tilde{\mathbf{x}}_0$ from the current partially masked state $\mathbf{x}_{t_k}$, then uses $\text{Remask}(\cdot)$ to fill in the masked positions with these predictions and re-apply masks to satisfy the next mask rate $t_{k-1}$.

## 2.2. From LLM-SAEs to DLM-SAEs: Training and Steering Differences

We summarize two key changes when adapting LLM-SAEs to DLMs: (i) which token positions provide training activations under diffusion corruption, and (ii) how feature steering is applied across denoising steps. Figure 2 shows training-position selection and per-step injection.

**Training difference.** In LLMs, SAE training typically collects activations from fully observed prefixes under a causal mask. In DLMs, each forward pass is conditioned on a random corruption level $t$ and a partially masked sequence $x_t$ (Eq. (3)), so we define:

$$\mathcal{M}(x_t) = \{ i \in [N] : x_t^i = \texttt{[MASK]} \},$$
$$\mathcal{U}(x_t) = [N] \setminus \mathcal{M}(x_t). \quad (5)$$

and let $x_\ell^i(x_t) \in \mathbb{R}^d$ be the residual-stream activation at layer $\ell$ and position $i$ when running the DLM on $x_t$. We train two DLM-SAEs by choosing the position set $\mathcal{S}(x_t) \in \{\mathcal{M}(x_t), \mathcal{U}(x_t)\}$ and minimizing

$$\mathbb{E}_{x_0 \sim q(x),\, t \sim U(0,1),\, x_t \sim q(x_t|x_0)}$$
$$\left[ \frac{1}{|\mathcal{S}(x_t)|} \sum_{i \in \mathcal{S}(x_t)} \left( \|x_\ell^i(x_t) - \hat{x}_\ell^i(x_t)\|_2^2 + \lambda \|h_\ell^i(x_t)\|_1 \right) \right]$$
$$(6)$$

where $(h_\ell^i, \hat{x}_\ell^i)$ follow Eq. (1) with shared SAE parameters at layer $\ell$. We refer to $\mathcal{S}(x_t) = \mathcal{M}(x_t)$ as MASK-SAE and $\mathcal{S}(x_t) = \mathcal{U}(x_t)$ as UNMASK-SAE.

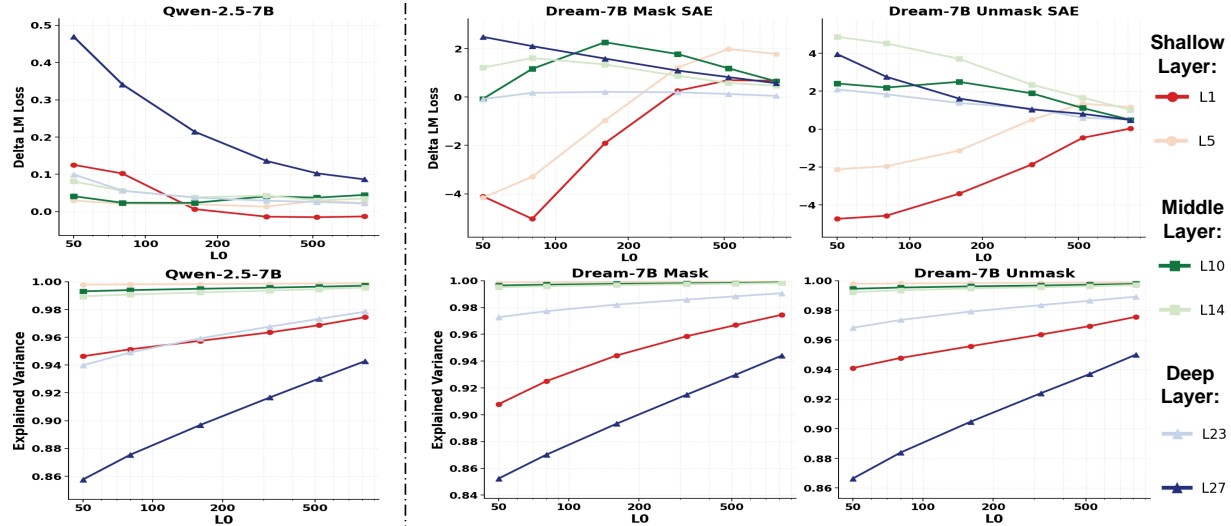

*Figure 3.* **Sparsity-fidelity trade-off for Qwen SAEs and Dream SAEs. Top row:** functional fidelity measured by $\Delta$LM loss (Eq. 10); **Bottom row:** reconstruction fidelity measured by explained variance (Eq. 9). **Columns:** a LLM baseline (Qwen-2.5-7B, left) versus Dream-7B SAEs (MASK-SAE, middle) and (UNMASK-SAE, right). This figure shows that Dream-SAEs achieve strong sparsity-fidelity trade-offs and even exhibit negative $\Delta$LM loss in shallow layers at small $L_0$, an effect absent or much weaker in the LLM baseline.

**Steering difference.** LLM steering applies a single intervention in a left-to-right pass. DLM steering must operate across denoising steps (Eq. (4)). Let $\mathbf{X}_{\ell,k} \in \mathbb{R}^{N \times d}$ denote the residual-stream matrix at layer $\ell$ when running on $\mathbf{x}_{t_k}$, and let $v_f \in \mathbb{R}^d$ be the decoder atom for feature $f$ (Eq. (2)). We inject the feature at step $k$ via:

$$\mathbf{X}_{\ell,k}^{\text{steer}} = \mathbf{X}_{\ell,k} + \alpha \, m_f \, \mathbf{s}_k \, v_f^\top, \qquad (7)$$

where $\mathbf{s}_k \in \{0,1\}^N$ selects token positions to steer at that step. We study two DLM-specific choices:

$$\begin{aligned} \text{ALL-TOKENS:} \quad & (\mathbf{s}_k)_i = 1 \; \forall i, \\ \text{UPDATE-TOKENS:} \quad & (\mathbf{s}_k)_i = \mathbf{1}[i \in \mathcal{M}(\mathbf{x}_{t_k})]. \end{aligned} \qquad (8)$$

Eq. (7) applies the same feature direction repeatedly over denoising, while Eq. (8) distinguishes steering all positions versus only the currently masked (to-be-updated) positions.

## 3. Can SAEs Extract Interpretable Features in DLMs?

In this section, we train SAEs and assess (i) whether they provide a faithful reparameterization of DLM activations under a controllable sparsity budget, and (ii) whether the resulting latents form human-interpretable features.

### 3.1. Training Details

We train Top-$K$ SAEs on two representative DLMs, Dream-7B (Ye et al., 2025) and LLaDA-8B (Nie et al., 2025b), using The Common Pile v0.1 (Kandpal et al., 2025) as training data. For each model, we splice the SAE into the residual

stream and train a SAE at layers spanning the network depth: two shallow, two middle, and two deep layers.

*Table 1.* Overview of DLM-SAE training configurations.

|  | **Dream-SAE** | **LLaDA-SAE** |
| --- | --- | --- |
| Model | Dream-7B | LLaDA-8B |
| Training Data | The Common Pile | The Common Pile |
| Activation function | ReLU+TopK | ReLU+TopK |
| Insertion layers | 1, 5, 10, 14, 23, 27 | 1, 6, 11, 16, 26, 30 |
| Insertion site | residual stream | residual stream |
| SAE width | 16K | 16K |

Table 1 summarizes the backbone-specific settings. Full hyperparameters, training resources and data preprocessing details are deferred to Appendix A.

### 3.2. Evaluate SAEs on sparsity-fidelity trade-off

We evaluate DLM-SAEs with two metrics: reconstruction fidelity and the change in DLM training loss when the SAE is spliced into the model. All metrics are computed on held-out tokens from the same distribution as training.

**Reconstruction fidelity (Explained Variance $\uparrow$).** We report explained variance (EV) of reconstructions, measuring how much of $x$ is captured by the reconstruction $\hat{x}$:

$$\text{EV} = 1 - \frac{\mathbb{E}\big[\|x - \hat{x}\|_2^2\big]}{\mathbb{E}\big[\|x\|_2^2\big]}. \qquad (9)$$

**Functional fidelity (delta LM loss $\downarrow$).** Let $\mathcal{L}_{\text{DLM}}(\theta)$ be the masked-token cross-entropy objective in Eq. (3). We define $\mathcal{L}_{\text{DLM}}^{\text{ins}}(\theta)$ as the same objective, but with the residual stream at the target layer replaced by the SAE reconstruction

*Table 2.* **SAE steering summary at** $L_0$=80 **across models and layers.** Each cell reports three quantities: $C$ (normalized concept improvement, larger is better), $P$ (normalized perplexity reduction, larger $P$ is better), and $S$ (overall steering score, $S = C + \lambda P$ with $\lambda$=0.3, larger is better). Stars ($\star$) mark the best $S$ within each model-family block (Qwen/Dream; LLaMA/LLaDA) at each layer. Overall, SAE features enable effective diffusion-time interventions that often outperform single-pass LLM steering.

| Model / SAE | Shallow Layer | | | | | | Middle Layer | | | | | | Deep Layer | | | | | |
| | L1 | | | L5 | | | L10 | | | L14 | | | L23 | | | L27 | | |
| | $C\uparrow$ | $P\uparrow$ | $S\uparrow$ | $C\uparrow$ | $P\uparrow$ | $S\uparrow$ | $C\uparrow$ | $P\uparrow$ | $S\uparrow$ | $C\uparrow$ | $P\uparrow$ | $S\uparrow$ | $C\uparrow$ | $P\uparrow$ | $S\uparrow$ | $C\uparrow$ | $P\uparrow$ | $S\uparrow$ |
|---|---|---|---|---|---|---|---|---|---|---|---|---|---|---|---|---|---|---|
| Qwen-2.5-7B | **0.16** | -0.39 | 0.04 | **0.18** | -0.34 | 0.08 | 0.18 | -0.35 | 0.08 | 0.22 | **-0.26** | **0.14**$^\star$ | 0.28 | -0.43 | 0.15 | 0.03 | -0.19 | -0.02 |
| Dream-Unmask | 0.13 | **-0.08** | 0.10 | 0.17 | -0.17 | **0.12**$^\star$ | **0.19** | -0.31 | 0.10 | 0.20 | -0.64 | 0.01 | 0.26 | **-0.33** | 0.16 | **0.33** | **0.18** | **0.38**$^\star$ |
| Dream-Mask | 0.14 | -0.11 | **0.11**$^\star$ | 0.13 | **-0.09** | 0.11 | 0.17 | **-0.12** | **0.13**$^\star$ | **0.22** | -0.37 | 0.11 | **0.35** | -0.35 | **0.24**$^\star$ | 0.29 | 0.01 | 0.29 |
| LLaMA-3-8B | 0.08 | -7.89 | -2.29 | **0.18** | -2.14 | -0.46 | **0.15** | -1.23 | -0.22 | **0.25** | -0.67 | 0.05 | 0.17 | -0.44 | 0.04 | 0.13 | -0.09 | 0.10 |
| LLaDA-Unmask | **0.14** | -0.05 | **0.13**$^\star$ | 0.16 | -0.13 | **0.13**$^\star$ | 0.13 | 0.02 | 0.13 | 0.13 | **-0.04** | 0.11 | **0.25** | -0.16 | **0.20**$^\star$ | **0.16** | -0.26 | 0.08 |
| LLaDA-Mask | 0.13 | **0.00** | **0.13**$^\star$ | 0.11 | **0.00** | 0.11 | 0.14 | **0.04** | **0.15**$^\star$ | 0.16 | -0.06 | **0.14**$^\star$ | 0.21 | **-0.12** | 0.17 | 0.15 | **-0.02** | **0.15**$^\star$ |

$(x \leftarrow \hat{x})$ before continuing the forward pass:

$$\Delta\mathcal{L}_{\mathrm{DLM}} = \mathcal{L}_{\mathrm{DLM}}^{\mathrm{ins}}(\theta) - \mathcal{L}_{\mathrm{DLM}}(\theta), \qquad (10)$$

computed under the same $(x_0, t, x_t)$ sampling as Eq. (3).

Figure 3 shows that DLM-SAEs achieve a favorable sparsity-fidelity profile, indicating that splicing SAEs are usable for mechanistic analyses. Interestingly, Dream shows a shallow-layer regime at small $L_0$ where $\Delta$LM loss is negative, meaning that SAE insertion can reduce cross-entropy loss. This effect is absent or much weaker in LLMs, where insertion increases loss. We observe the same pattern on LLaDA-SAEs, with full results in Appendix B.

### 3.3. Interpretability of Features

To test whether DLM-SAEs learn human-interpretable concepts, we adopt auto-interpretation protocol (Karvonen et al., 2025; Paulo et al., 2025) that produces (i) explanation for each feature and (ii) interpretability score measuring how predictive that explanation is of feature activation.

For each feature $f$, we apply an automated interpretation procedure on a held-out stream of 5M tokens. We highlight the tokens that activate $f$ the most (along with their activation values) and ask an LLM to describe the pattern that $f$ appears to capture. To check whether this description is meaningful, we run a simple discrimination test: given unlabeled sequences, a separate judge LLM uses the description to predict which ones should activate $f$, and we report its accuracy as the interpretability score. Full details are in Appendix C, with examples in Appendix D.

**In conclusion, SAEs serve as a practical mechanistic interpretability interface for DLMs.** The resulting DLM-SAEs achieve strong sparsity-fidelity trade-offs, including an early-layer regime where SAE insertion can even reduce cross-entropy loss and yield human-interpretable features validated by automated interpretation scores.

## 4. Do These Features Enable Effective Steering During Denoising Stage?

### 4.1. Using Features to Intervene during Denoising

A key advantage of DLMs is that generation unfolds over multiple denoising steps, exposing repeated opportunities to intervene. We continue the idea of using SAE features for steering in LLMs: injecting its decoder atom during denoising to change the final decoded text.

Concretely, we intervene at a chosen layer $\ell$ by adding the feature direction $v_f$ to the residual stream according to Eq. (7). We consider two diffusion-time steering strategies that differ only by the token selector $\mathbf{s}_k$ in Eq. (7): ALL-TOKENS steers all positions, while UPDATE-TOKENS steers only the currently masked positions $\mathcal{M}(\mathbf{x}_{t_k})$ (Eq. (8)).

### 4.2. Steering Results

**Metrics.** We follow the steering evaluation protocol of (Sun et al., 2025; Wu et al., 2025). For each feature $f$, we sample neutral prefixes $\mathcal{P}$ (examples in Appendix E) and generate continuations with and without diffusion-time steering (Eq. 7). Let $C_{\mathrm{before}}(f), C_{\mathrm{after}}(f)$ denote the prefix-averaged concept scores, and let $p_{\mathrm{before}}(f), p_{\mathrm{after}}(f)$ denote the prefix-averaged perplexities, both computed on the generated continuation. We report three normalized metrics:

$$C(f) = \frac{C_{\mathrm{after}}(f) - C_{\mathrm{before}}(f)}{s_C} \quad (s_C \in \{1, 100\}), \tag{11}$$

where $C(f)$ measures the concept improvement (larger is better; typically $C(f) \in [-1, 1]$ after normalization).

$$P(f) = \frac{p_{\mathrm{before}}(f) - p_{\mathrm{after}}(f)}{p_{\mathrm{before}}(f)}, \tag{12}$$

where $P(f)$ is the relative perplexity reduction (larger is better; $P(f) \leq 1$ and can be negative if fluency degrades).

$$S(f) = C(f) + \lambda P(f), \tag{13}$$

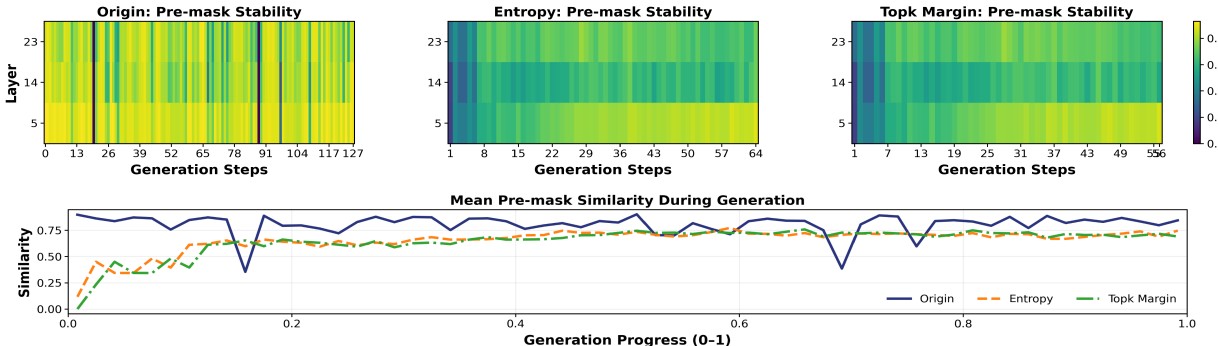

*Figure 4.* **Pre-mask SAE feature stability across three DLM inference orders.** Columns compare three different decoding orders. Each step on GSM8K rollouts. **Top:** layer-step heatmaps of mean pre-mask top-$k$ Jaccard similarity between consecutive steps ($k-1 \rightarrow k$). **Bottom:** mean pre-mask similarity (averaged over tracked layers/positions) vs. normalized generation progress. This figure shows that ORIGIN yields a less dynamic SAE trajectory, while confidence-based orders exhibit structured turnover followed by stabilization.

where $S(f)$ is the overall steering score trading off concept gain and fluency ($S(f) \in [-1, 1+\lambda]$).

**Experimental setup.** We evaluate diffusion-time steering for 30 denoising steps. For Dream-7B we use ALL-TOKENS steering, i.e., $(\mathbf{s}_k)_i = 1$ for all positions (Eq. 8), while for LLaDA-8B we use UPDATE-TOKENS steering, i.e., $(\mathbf{s}_k)_i = \mathbf{1}[i \in \mathcal{M}(\mathbf{x}_{t_k})]$. For fairness, we match the generation length budget and the steering-strength sweep across LLM and DLM baselines by using the same $\alpha$ range. Full details (steering settings, prompt set, ablation experiments and feature selection) are deferred to Appendix F.

Table 2 shows a clear trend: **sparse features enable diffusion-time interventions that are effective and often outperform LLM steering.** Relative to LLM-SAEs, DLM-SAEs achieve substantially higher overall steering scores, typically by 2–10× in deep layers. The strongest gains appear in the deepest layers (e.g., L27), where DLM steering attains the best steering score within each model family, suggesting that steerable semantic (concept-level) directions are concentrated in late residual-stream representations. Overall, diffusion-time steering offers a more effective control interface than single-pass LLM steering.

## 5. Can SAEs Help Explain Different Decoding Order of DLMs?

In this section, we use SAEs to track residual-stream dynamics and analyze how representation trajectories differ across decoding-order strategies.

### 5.1. Analyzing Decoding-Order Strategies with Features

We consider three remasking strategies that control the token generation order: (i) ORIGIN (Austin et al., 2021) updates positions in a random order, independent of model confidence; (ii) TOPK-MARGIN (Kim et al., 2025) ranks

positions by the margin score $p_{(1)} - p_{(2)}$ and updates the top-$K$ most confident positions first; (iii) ENTROPY (Ye et al., 2025) ranks positions by token-distribution entropy and updates lower-entropy positions earlier.

For a decoding order $\mathcal{O}$, let $\mathbf{x}_{t_k}^{(\mathcal{O})}$ denote the partially masked sequence at denoising step $k$, and let $h_{\ell,k,i}^{(\mathcal{O})}$ be the SAE latent at layer $\ell$ and position $i$ obtained by encoding the corresponding residual-stream activation (Eq. (1)). Let $\mathrm{TopK}_{\mathrm{feat}}(h)$ return the indices of the $K_{\mathrm{feat}}$ largest-magnitude latents. Feature-set overlap is measured with Jaccard similarity (Jaccard, 1912), $J(A, B) = \frac{|A \cap B|}{|A \cup B|}$.

**Pre-mask stability.** To test whether a position becomes stable before it is decoded, we measure step-to-step similarity while the position remains masked. We define: $\mathcal{M}(\mathbf{x}_{t_k}^{(\mathcal{O})}) = \{ i : \mathbf{x}_{t_k}^{(\mathcal{O}),i} = \mathtt{[MASK]} \}$. For $i \in \mathcal{M}(\mathbf{x}_{t_k}^{(\mathcal{O})})$:

$$S_{\ell,k,i}^{\mathrm{pre}}(\mathcal{O}) = J\Big(\mathrm{TopK}_{\mathrm{feat}}(h_{\ell,k,i}^{(\mathcal{O})}), \mathrm{TopK}_{\mathrm{feat}}(h_{\ell,k-1,i}^{(\mathcal{O})})\Big) \tag{14}$$

We summarize $S_{\ell,k,i}^{\mathrm{pre}}(\mathcal{O})$ by averaging over masked positions, prompts, and runs, and visualize it as a function of layer and generation progress.

**Post-decode drift.** To measure how much a decoded position's representation continues to change, define its decode step $k_i(\mathcal{O}) = \min\{k : i \notin \mathcal{M}(\mathbf{x}_{t_k}^{(\mathcal{O})})\}$. Let $K$ be the total number of denoising steps. We quantify average drift as:

$$D_{\ell,i}^{\mathrm{post}}(\mathcal{O}) = \frac{1}{K - k_i(\mathcal{O})} \sum_{k=k_i(\mathcal{O})+1}^{K} \Big(1 - S_{\ell,k,i}(\mathcal{O})\Big) \tag{15}$$

To compare strategies with different step counts, we plot summaries against normalized progress $\tau = k/K \in [0, 1]$.

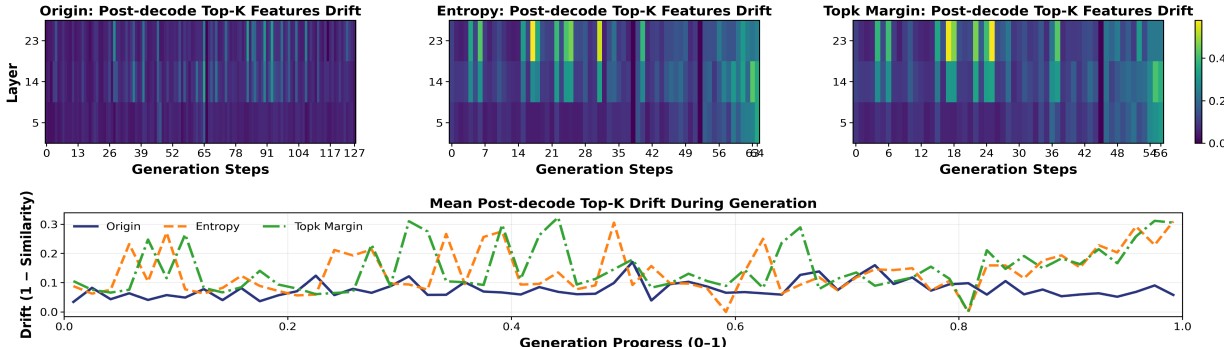

*Figure 5.* **Post-decode SAE feature drift.** Drift is computed only after a position's token is fixed, highlighting semantic changes. **Top:** layer-step heatmaps of post-decode top-$k$ drift between consecutive steps' top-$k$ feature sets. **Bottom:** mean post-decode drift (averaged over tracked positions) vs. normalized generation progress. This figure shows that confidence-based orders sustain stronger deep-layer post-decode drift, indicating the effect of bidirectional attention is stronger in this situation, whereas ORIGIN drifts less.

## 5.2. Feature Dynamics Across Decoding Strategies

We study decoding-order effects on GSM8K (Cobbe et al., 2021) using Dream-7B. To ensure coverage and reduce prompt-specific artifacts, we sample 800 questions and run diffusion inference with $K{=}128$ denoising steps, generating one token per step. In this setting, the three decoding orders yield markedly different task performance: ORIGIN achieves $8\%$ accuracy, while the confidence-based orders perform substantially better (TOPK-MARGIN: $56\%$, EN-TROPY: $59\%$). We track SAE latents on the same set of layers across orders and compute pre-mask stability $S^{\mathrm{pre}}$ (Eq. 14) and post-decode drift $D^{\mathrm{post}}$ (Eq. 15).

Figures 4 and 5 show a clear difference between decoding orders. Under ORIGIN, the SAE features change only slightly from step to step, suggesting a quieter evolution in the SAE feature space. In contrast, the confidence-based orders (TOPK-MARGIN, ENTROPY) show a more organized progression: features shift more early on for still-masked tokens, then settle quickly, while deep layer features continue to adjust after tokens are fixed, indicating the effect of bidirectional attention is stronger in this situation. Further analysis can be found in Appendix G.

**We conjecture that these SAE-based dynamics provide a useful signal that correlates with task performance across decoding orders.** In our experiment, the higher accuracy confidence-based orders (TOPK-MARGIN, ENTROPY) show stronger early changes on masked positions and continued deep-layer adjustment after decoding, whereas the lower-performing ORIGIN order changes less throughout. Notably, we observe systematic layerwise differences in decoding-order dynamics between higher and lower performing strategies in the SAE feature space, offering mechanistic insight to guide future decoding-order design.

## 6. Do Base-Trained SAEs Generalize to Instruction-Tuned Model?

### 6.1. Layerwise Transfer: Base SAE vs. SFT SAE

Training SAEs on every instruction-tuned DLM is expensive. Ideally, a base-trained SAE could be reused as a faithful one on the instruction-tuned model. We therefore test whether Dream base SAEs and Dream SFT SAEs behave similarly when inserted into the same backbone.

We compare two SAEs trained with identical settings: BASE SAE, trained on Dream-7B Base (DREAM BASE), and SFT SAE, trained on Dream-7B Instruct (DREAM SFT). For each target model, we splice the SAE into the residual stream at layers and sweep sparsity budgets. We evaluate (i) reconstruction quality using explained variance (Eq. (9)) and (ii) functional faithfulness using the insertion-induced loss change $\Delta\mathcal{L}_{\mathrm{DLM}}$ (Eq. (10)). Results for inserting SAEs into the (DREAM BASE) are deferred to Appendix H.

The left and middle panels of Figure 6 show that except at the deepest layer, inserting BASE SAE or SFT SAE into the DREAM SFT yields nearly identical functional behavior. Across L1, L5, L10, L14, and L23, the experiments consistently indicate strong cross-model transferability of SAE representations at these depths. By contrast, L27 exhibits a clear separation in $\Delta\mathcal{L}_{\mathrm{DLM}}$, suggesting that the deepest representations are substantially more sensitive to instruction tuning and alignment effects.

### 6.2. DLM-SAE Transfer Under Instruction Rollouts

We further test transfer under instruction rollouts to assess whether Base-SFT SAE transfer persists on instruction data. Concretely, we generate DREAM SFT assistant continuations with 30 steps and 30 tokens, insert an SAE at layers $\{1, 5, 10, 14, 23, 27\}$. Insertion-induced functional change is measured by $\Delta\mathcal{L}_{\mathrm{DLM}}$ (Eq. (10)) on the rollout segment.

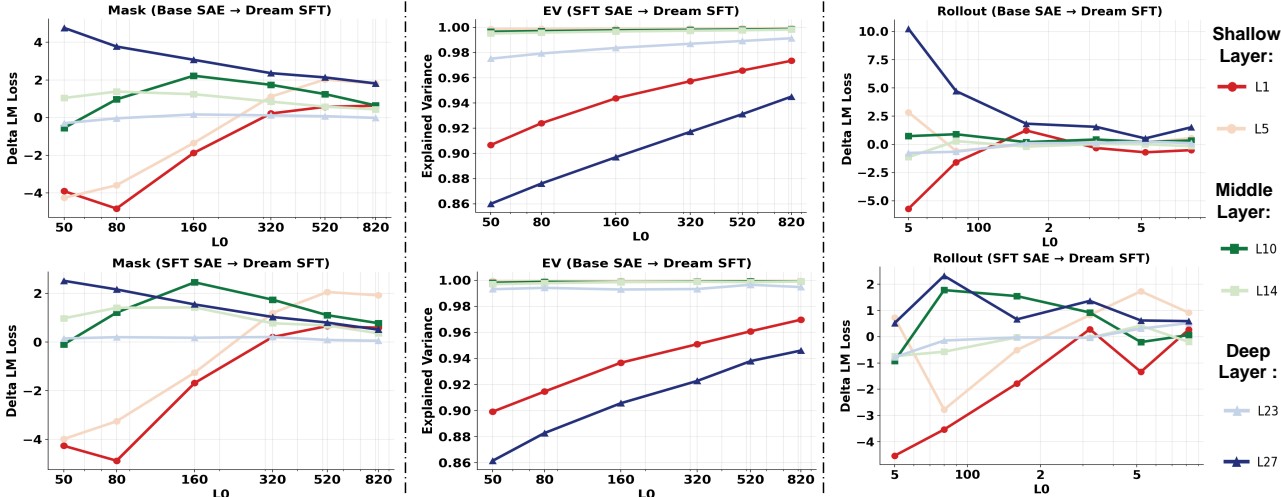

*Figure 6.* **Base-to-SFT transfer of DLM-SAEs across layers.** We chose MASK-SAE to test generalization. **Top row:** BASE SAE inserted into DREAM SFT. **Bottom row:** SFT SAE inserted into DREAM SFT. **Left:** $\Delta\mathcal{L}_{\text{DLM}}$ evaluated on masked-token denoising inputs. **Middle:** reconstruction fidelity measured by Explained Variance (EV). **Right:** $\Delta\mathcal{L}_{\text{DLM}}$ measured during instruction rollouts. This figure indicates Base-trained SAEs transfer nearly losslessly to instruction-tuned DLM except at the deepest layer.

Despite distribution shift, SAEs remain reusable in shallow and middle layers on instruction rollouts (the right part of Figure 6). However, in Layer 27, BASE SAE induces a much larger loss increase than SFT SAE. This suggests that the base model's deepest-layer subspace fails to capture instruction-critical directions engaged during rollouts.

**Base-trained SAEs generally transfer almost losslessly to the instruction-tuned DLM, except at the deepest layer.** We can see BASE SAE and SFT SAE behave almost identically on DREAM SFT for L1-L23, suggesting highly similar internal signals at these depths. This reuse largely holds under instruction rollouts, but breaks at L27, where the BASE SAE fails to capture tuning-specific behavior.

## 7. Related Work

**SAEs as interpretability interface for LLMs** Recent work has already established highly practical SAE interfaces for LLMs (Yang et al., 2024; Grattafiori et al., 2024; McDougall et al.; He et al., 2024), training on residual-stream activations to learn feature dictionaries that reconstruct faithfully and capture meaningful features (Mudide et al., 2025). After obtaining these interpretable features, the first clear practical payoff has been *steering*: intervening along a direction in the residual stream to shape model behavior (Subramani et al., 2022; Rimsky et al., 2024; Turner et al., 2024; Stolfo et al., 2025). Prior steering methods often search directly in raw hidden state space, where directions can be semantically polysemantic, limiting interpretability (Mayne et al., 2024; Arad et al., 2025; Wang et al.). In contrast, SAE feature directions provide more "atomic" control (O'Brien et al., 2025; Zhao et al., 2025; Farrell et al., 2024; Wang et al.,

2025b). *In contrast to this pipeline, we present DLM-Scope, the first SAE-based interpretability interface for DLMs.* Empirically, we uncover DLM-specific behavior (e.g., inserting an SAE into early layers can reduce cross-entropy loss, a phenomenon that is absent in LLMs).

**The rise path of DLMs** Diffusion language models have recently become increasingly competitive for text understanding and generation (Chen et al., 2025; Li et al., 2022; DeepMind, 2024; Zhang et al., 2025; Wang et al., 2025a). In this context, a range of high-performing DLMs has emerged along two lines: continuous diffusion language models (Ho et al., 2020; Liu et al., 2022) and discrete diffusion language models (He et al., 2023; Fan et al., 2026). Prior work has developed interpretability tools for text-to-image diffusion models: for example, SAEs edits expose sparse, causal concept factors that can be amplified or erased with minimal collateral effects (Surkov et al., 2025). Complementary work uses attention and causal localization to link tokens to image regions and to localize and edit specific visual knowledge efficiently. (Tang et al., 2023; Helbling et al., 2025; Basu et al., 2023; Shi et al., 2025). By comparison, DLMs still lack interpretability interface and need the corresponding tools. *Our work train SAEs for DLMs and shows that they enable diffusion-time steering, decoding-strategy analysis, and broad transfer to instruction-tuned model.*

## 8. Discussion

This work presents **DLM-Scope**, the first SAE-based interpretability interface for diffusion language models (DLMs). We highlight two points: **(i)** SAEs trained on DLM denoising activations are usable for mechanistic analyses, and SAE

insertion can even reduce cross-entropy loss in certain layers, a behavior distinct from LLMs. **(ii)** DLM-SAEs yield actionable sparse features that enable effective diffusion-time interventions (often outperforming LLM steering), support decoding-strategy analysis, and transfer broadly to instruction-tuned DLM. Looking ahead, as DLMs continue to scale and deploy across more settings, interpretability tools will become increasingly critical. Future work will train and apply SAEs across a broader range of DLMs to better understand and improve diffusion-time behavior.

## Acknowledgements

We would like to thank the anonymous reviewers and area chairs for their helpful comments. We acknowledge the support from NSFC 62306252, Hong Kong ECS award 27309624, Guangdong NSF 2024A1515012444, and the central fund from HKU.

## Impact Statement

We introduce **DLM-Scope**, the first sparse autoencoder (SAE) interpretability interface designed for diffusion language models (DLMs). By training SAEs on DLMs and defining DLM-specific training and denoising-time intervention choices, our framework enables feature-level analysis of DLM internals, systematic evaluation under sparsity constraints, and controllable diffusion-time steering. We expect the primary positive impact to be improved transparency, debugging, and safety auditing of modern generative systems. Our results show that SAEs can extract human-interpretable features in DLMs and provide a practical lens to quantify how semantic representations evolve across denoising steps. As DLMs become more widely adopted, we anticipate this kind of interpretability and intervention tooling will be increasingly valuable for understanding model behavior and supporting responsible development.

As with many interpretability and control tools, the same interface may be used to steer model behavior more effectively, which can be beneficial but could also be misused. We provide diagnostic and controllable tooling for existing DLMs. We therefore do not anticipate risks beyond those commonly associated with interpretability research, while encouraging careful evaluation and reporting of intervention settings and side effects.

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

# A. DLM-SAE Training Setup

We train Top-$K$ sparse autoencoders (TopK-SAEs (Gao et al., 2024; Team, 2024)) on residual-stream activations at selected layers to learn a sparse dictionary over the layerwise activation distribution. Concretely, for a layer activation tensor of shape $(B, T, d)$, we treat each valid token position as one training example by flattening $(B, T, d) \rightarrow (B \cdot T, d)$ and training the SAE on the resulting per-token vectors. This design aligns with the objective of modeling the full distribution of internal representations at a layer and improves sample efficiency compared to restricting training to any single position.

## A.1. Training Hyperparameters

Across backbones, we use a fixed context length of 2048 and stream tokenized text from a Common Pile split (Kandpal et al., 2025). A typical quick-check setting trains with batch size 8 under a 1M-token budget, and the same pipeline scales to larger budgets. For diffusion language models, inputs are partially masked, so we train two controlled variants under the same TopK-SAE objective: MASK-SAE collects activations only at masked (to-be-predicted) positions, while UNMASK-SAE collects activations only at unmasked context positions. This isolates how the diffusion corruption pattern affects the learned sparse dictionary.

## A.2. Setup

We train a grid of SAEs per backbone spanning multiple layers and sparsity budgets; a typical setup trains 36 SAEs per model, corresponding to 6 layers × 6 sparsity settings. Table 3 summarizes representative training configurations and resource footprints across backbones, including the token budget, number of SAEs, SAE architecture, per-SAE storage size, per-SAE wall-clock time, batch/context settings, and hardware.

*Table 3.* Training configuration and resource footprint summary across model backbones, including token budget, number of SAEs, SAE architecture, per-SAE storage size, batch and context settings, and hardware.

| Model | Tokens | #SAEs | Arch | Disk/SAE | batch_size | context_length | sae_batch_size | GPU |
|-------|--------|-------|------|----------|------------|----------------|----------------|-----|
| Qwen2.5-7B | 150M | 36 | TopK | 449M | 8 | 2048 | 2048 | 2×A800 |
| Dream-7B | 150M | 36 | TopK | 449M | 8 | 2048 | 2048 | 2×A800 |
| LLaMA-8B | 150M | 36 | TopK | 512M | 8 | 2048 | 2048 | 1×H100 |
| LLaDA-8B | 150M | 36 | TopK | 512M | 8 | 2048 | 2048 | 6×H100 |

Table 3 indicates that we keep the batch and context settings fixed across backbones to facilitate fair comparisons, while the observed per-SAE training time and hardware requirements vary with model family and experimental variant. In addition, the per-SAE checkpoint size is dominated by the SAE width and decoder parameters, leading to comparable storage footprints within each hidden-size family.

# B. Full Sparsity-Fidelity Results for LLaDA-SAEs with Explained Variance and $\Delta$LM Loss

This appendix reports full layerwise sparsity-fidelity sweeps for LLaDA-SAEs using the same evaluation metrics as the main text: reconstruction fidelity via explained variance (Eq. (9)) and functional fidelity via insertion-induced loss change $\Delta\mathcal{L}_{\text{DLM}}$ (Eq. (10)). Evaluation follows the same diffusion-style masking protocol as in the main experiments.

**Functional fidelity across layers and $L_0$.** Figure 7 plots $\Delta\mathcal{L}_{\text{DLM}}$ over an $L_0$ sweep across representative insertion layers. Increasing $L_0$ generally reduces the magnitude of insertion-induced loss change (curves move toward 0), while shallow-layer regimes can exhibit negative $\Delta\mathcal{L}_{\text{DLM}}$, indicating that SAE insertion may improve the denoising objective. In deeper layers, overly sparse SAEs tend to induce larger functional deviations.

**Reconstruction fidelity across layers and $L_0$.** Figure 8 plots explained variance (EV) versus $L_0$ across layers. EV increases monotonically with $L_0$, with shallow and mid layers achieving high reconstruction fidelity at moderate sparsity budgets, while deeper layers typically require larger $L_0$ to reach comparable fidelity.

**Consistency with Dream.** Overall, the LLaDA results mirror the main-text Dream findings: EV improves steadily as sparsity is relaxed, and deeper layers are more sensitive when SAEs are too sparse. Inserting SAE can even reduce the

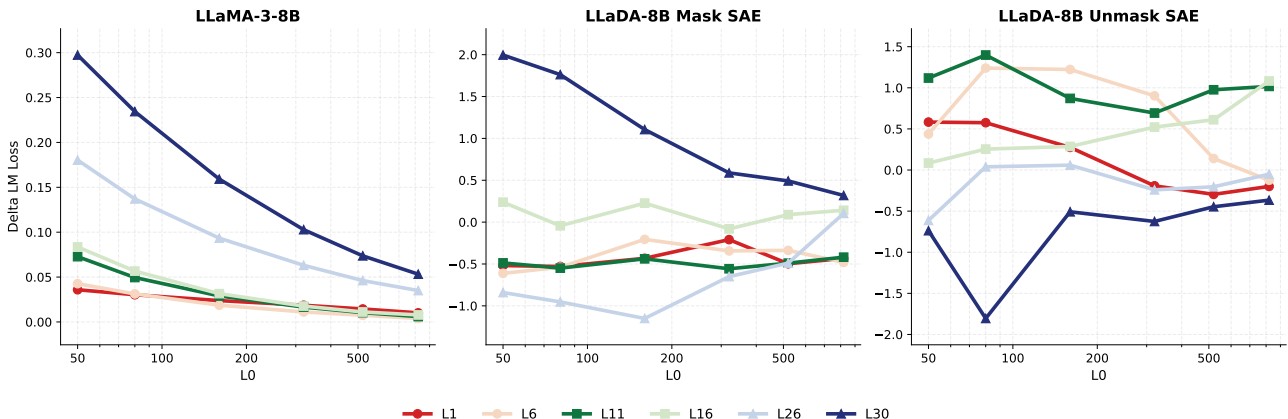

Figure 7. **Full $\Delta\mathcal{L}_{\mathrm{DLM}}$ sparsity sweeps for LLaDA-SAEs.** Each curve corresponds to inserting an SAE at a specific residual-stream layer (legend) while sweeping the sparsity budget $L_0$. The vertical axis reports the insertion-induced denoising loss change $\Delta\mathcal{L}_{\mathrm{DLM}}$ (Eq. (10)), evaluated on held-out masked-token denoising inputs using the same corruption/inference setup as in the main experiments. More negative values indicate that inserting the SAE improves the DLM denoising objective, whereas positive values indicate a degradation. Plotting all layers together makes it easy to compare how functional impact varies with sparsity and depth under matched evaluation conditions.

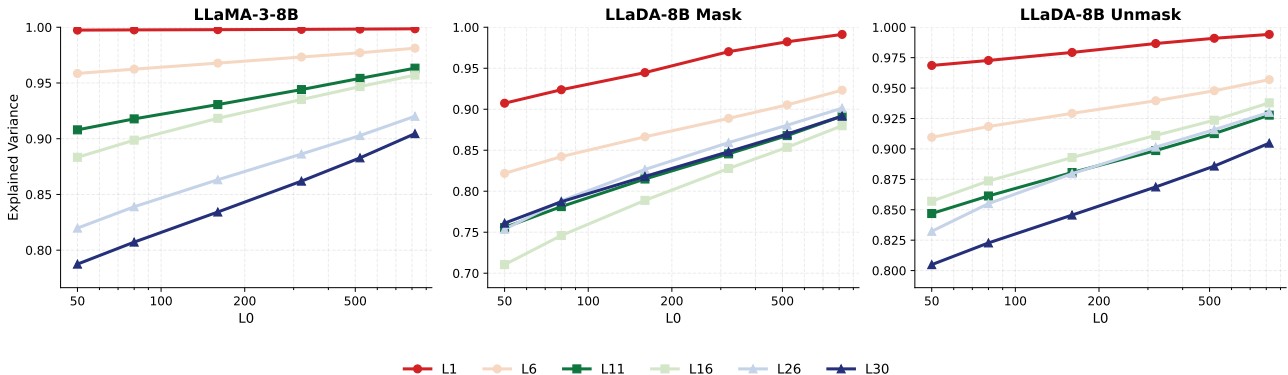

Figure 8. **Full explained-variance sparsity sweeps for LLaDA-SAEs.** Each curve corresponds to inserting an SAE at a specific residual-stream layer (legend), while sweeping the sparsity budget $L_0$ on the horizontal axis. The vertical axis reports explained variance (EV; Eq. (9)), computed on held-out activations, and reflects how well the SAE reconstruction captures variance in the target-layer representation under the chosen sparsity constraint. Viewing all layers on the same plot highlights how reconstruction fidelity scales with $L_0$ and how the EV-sparsity trade-off differs across depths.

cross-entropy loss in DLMs at some layers.

## C. Auto-Interpretation Protocol for SAE Features with Prompting and Scoring

We follow an auto-interpretation protocol (Paulo et al., 2025; Karvonen et al., 2025) that assigns each SAE latent a short natural-language description together with an interpretability score.

For each evaluated latent $f$, we build evidence windows from held-out text: (i) top-activating context windows around the strongest activation peaks, and (ii) additional importance-weighted windows from the non-top region, augmented with randomly sampled negative windows.

A judge LLM is queried in two stages. It first produces a concise explanation from the marked high-activation examples, then predicts which unmarked scoring examples should activate given that explanation. The auto-interpretability score is the agreement accuracy between the judge-selected indices and ground-truth active labels induced by the activation threshold on the same windows.

**Experimental configuration.** In our Dream-7B feature interpretation runs, we use Common Pile split and collect 5M tokens.

We tokenize into fixed windows of `context_length`=128 and run batched forward passes with `batch_size`=64. We evaluate up to `n_latents`=1000 latents per SAE, with `latent_batch_size`=100 controlling how many latents are processed per scheduling batch, and we filter dead latents using `dead_latent_threshold`=15 (minimum estimated activation count over the token budget). The judge model is `gpt-4o-mini`.

---

**Explanation prompt (generation stage)**

**System.**     We're studying neurons in a neural network.  Each neuron activates on some particular word/words/substring/concept in a short document.  The activating words in each document are indicated with << ... >>.  We will give you a list of documents on which the neuron activates, in order from most strongly activating to least strongly activating.  Look at the parts of the document the neuron activates for and summarize in a single sentence what the neuron is activating on.  Try not to be overly specific in your explanation.  Note that some neurons will activate only on specific words or substrings, but others will activate on most/all words in a sentence provided that sentence contains some particular concept.  Your explanation should cover most or all activating words.  Pay attention to capitalization and punctuation, since they might matter.

**User (template).** The activating documents are given below: 1. ... 2. ... ... N. ...

> **Input formatting.** Examples are detokenized windows ranked by activation strength; the putative activating span is marked with << >>.

---

**Scoring prompt (prediction stage)**

**System.**     We're studying neurons in a neural network.  Each neuron activates on some particular word/words/substring/concept in a short document.  You will be given a short explanation of what this neuron activates for, and then be shown several example sequences in random order.  You must return a comma-separated list of the examples where you think the neuron should activate at least once, on ANY of the words or substrings in the document.  For example, your response might look like "2, 9, 10, 12".  Try not to be overly specific in your interpretation of the explanation.  If you think there are no examples where the neuron will activate, you should just respond with "None".  You should include nothing else in your response other than comma-separated numbers or the word "None" – this is important.

**User (template).**     Here is the explanation: *<one-sentence explanation>*.  Here are the examples: 1. ... 2. ... ... N. ...

> **Output constraint.** The judge must output only a comma-separated list of indices (1-based) or `None`; scoring examples are shown without << >> markers.

---

## D. Examples of DLM-SAE Features with Maximally Activating Contexts and Explanations

We presents qualitative examples of DLM-SAE features discovered by the auto-interpretation pipeline. For each feature, we report its latent ID, the judge-produced explanation, the auto-interpretation scoring outcome (predicted active indices vs. ground-truth active indices, and the resulting accuracy), and one maximally activating context window (the highest-activation example from the generation set). These examples illustrate both highly precise substring-level features and broader concept-level features.

---

**Feature 37: ``ht'' substring detector**

**Explanation.** The neuron responds to the two-letter sequence ``ht'' (case-insensitive), appearing in acronyms and tokens such as `HTS`, `CFHTLS`, `DHT-11`, `\mathtt`, and names like `Mehta`.

**Auto-interpretation scoring.** Predicted active indices: $\{1, 4, 7, 9\}$; ground-truth active indices: $\{1, 4, 7, 9\}$; accuracy $= 1.00$.

**Maximally activating context.** `Bulk high-temperature superconductors (<<HT>><>) are capable of generating very strong magnetic fields`

---

---

**Feature 20: single-letter LaTeX variable token**

**Explanation.** Activates on single-letter mathematical variable tokens in LaTeX math, especially `k` in subscripts/superscripts, and similar single-letter variables (e.g., `n`, `q`) in math expressions.

**Auto-interpretation scoring.** Predicted active indices: $\{1, 4, 7, 9, 10, 13\}$; ground-truth active indices: $\{2, 4, 7, 13\}$; accuracy $= 0.714$.

**Maximally activating context.** `over $\mathbb{P}r̂_{<<k>>} \times \mathbb{P}$`

---

**Feature 2621: Internet of Things concept**

**Explanation.** Activates on mentions of the Internet of Things concept and related tokens, including ``Internet'', ``of'', ``Things'', ``IoT'', and references to connected devices.

**Auto-interpretation scoring.** Predicted active indices: $\{5, 8, 14\}$; ground-truth active indices: $\{5, 6, 8, 14\}$; accuracy $= 0.929$.

**Maximally activating context.** `[Max-activation example omitted here; insert the top activating window for Feature 2621.]`

---

**Feature 113: character-level ``e'' sensitivity**

**Explanation.** The neuron responds to the lowercase letter `e`, either as a standalone symbol (e.g., $e^+/e^-$ in physics or scientific notation markers) or embedded within words.

**Auto-interpretation scoring.** Predicted active indices: $\{1, 2, \ldots, 14\}$; ground-truth active indices: $\{3, 5, 9, 14\}$; accuracy $= 0.286$.

**Maximally activating context.** `prediction performance compared to all single-modality sing<<e>>-task baselines AUROC = 0.`

## E. Neutral Prefix Set for Steering Evaluation with Full Prompt List

To evaluate diffusion-time steering, we compare generations produced with and without feature injection under otherwise identical decoding settings. A practical challenge is that prompts themselves can strongly bias topic, sentiment, and style, which can confound both (i) measured concept changes and (ii) fluency metrics such as perplexity. To reduce this prompt-induced variance, we follow the neutral-prefix design philosophy used in prior steering evaluations, in which short, generic prefixes are sampled to provide minimal semantic constraints while still anchoring generation to a natural continuation format (Wu et al., 2025). Concretely, for each feature we sample multiple neutral prefixes and average the resulting concept and fluency scores across prefixes, yielding a more stable estimate of steering effectiveness that is less sensitive to any single prompt phrasing.

---

**Neutral prefix pool**

```
"Findings show that", "I once heard that", "Then the man said:", "I believe that", "The
news mentioned","She saw a","It is observed that","Studies indicate that","According to
reports,", "Research suggests that", "It has been noted that", "I remember when", "It all
started when", "The legend goes that", "If I recall correctly,", "People often say that",
"Once upon a time,", "It's no surprise that", "Have you ever noticed that", "I couldn't
believe when", "The first thing I heard was", "Let me tell you a story about", "Someone
once told me that", "It might sound strange, but", "They always warned me that", "Nobody
expected that","Funny thing is,","I never thought I'd say this, but","What surprised me
most was", "The other day, I overheard that", "Back in the day,", "You won't believe what
happened when", "A friend of mine once said,", "I just found out that", "It's been a long
time since", "In my experience,", "The craziest part was when", "If you think about it,",
"I was shocked to learn that","For some reason,","I can't help but wonder if","It makes
sense that", "At first, I didn't believe that", "That reminds me of the time when", "It
all comes down to", "One time, I saw that", "I was just thinking about how", "Imagine a
world where","They never expected that","I always knew that"
```

---

# F. Diffusion-Time Steering Experimental Details and Ablation Experiments

## F.1. Steering Setup

Dream generation uses `diffusion_generate` with `dlm_steps` denoising steps, iteratively refining a full-sequence state and producing up to `max_new_tokens` new tokens. Once enabled, the intervention is applied at every denoising step; we refer to Eq. (7) and Eq. (8) for the formal definition.

We use two position-selection families consistent with Eq. (8). `token_scope=all` implements ALL-TOKENS steering by injecting at every position each step. `token_scope=topk_tokens` is a step-dependent sparse selector that recomputes the top-$K$ activated positions at each step and injects only there, aligning with the same design class as UPDATE-TOKENS. With bidirectional attention and full-sequence denoising, effects can propagate globally across the sequence.

Steering evaluation uses `n_prefix`=5 neutral prefixes (Appendix E) and generates `max_new_tokens`=30 tokens. We report Concept improvement, relative perplexity change, and the combined steering score as defined in the main text, and sweep sparsity budgets $L_0$ when comparing SAEs.

Figure 9 and Figure 10 visualize the layer-wise steering results summarized in Table 2, plotting Concept improvement, relative perplexity change, and the combined steering score against the sparsity budget $L_0$ (Visualization of Table 2).

*Figure 9.* **Steering metrics vs. $L_0$ for Qwen-2.5-7B and Dream-7B SAEs** The figure contains three columns: Concept improvement (left), relative perplexity change (middle), and steering score (right).

Across models and layers, these visualizations provide a direct view of the same results as Table 2, highlighting how steering outcomes vary with $L_0$ and layer depth under matched evaluation settings.

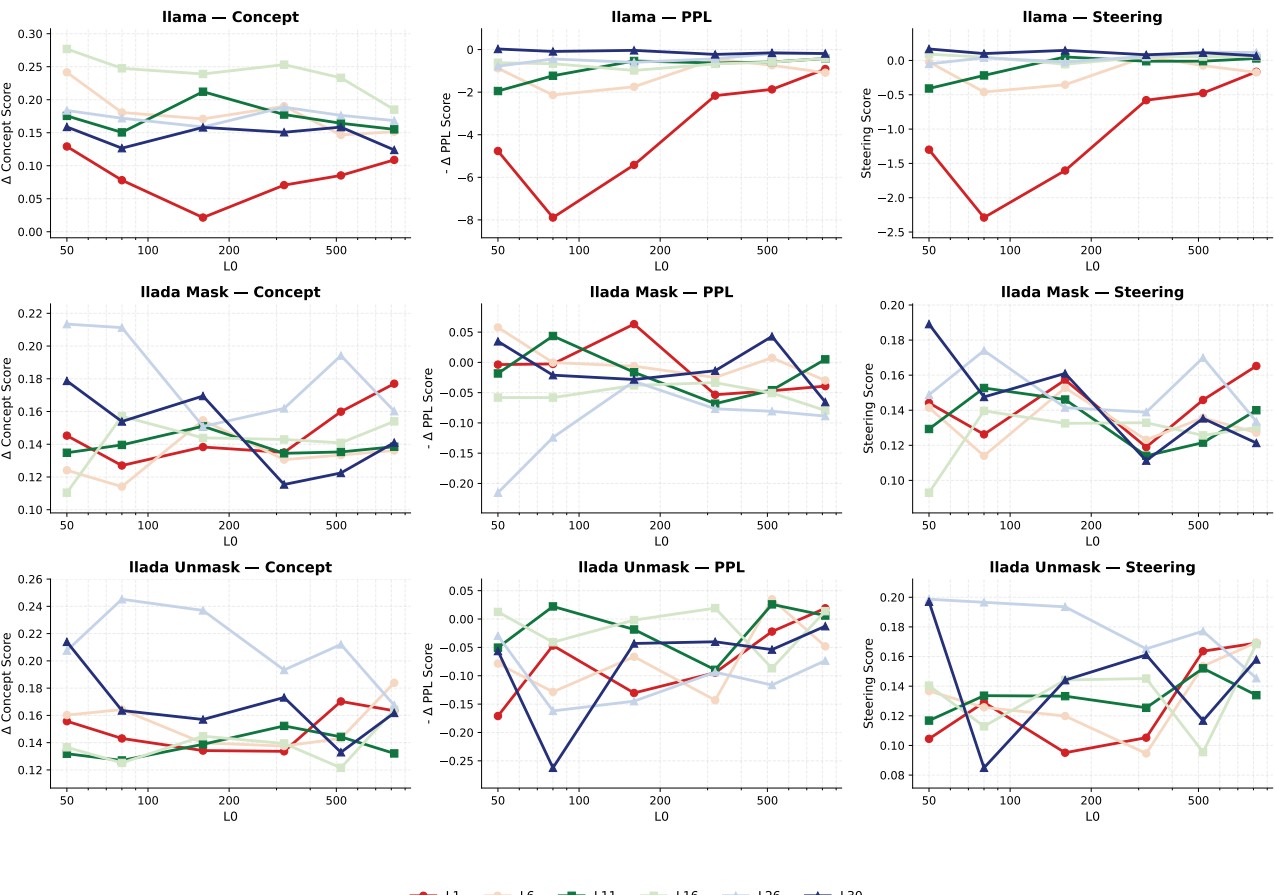

*Figure 10.* **Steering metrics vs. $L_0$ for LLaMA-8B and LLaDA-8B SAEs.** Each row corresponds to a target layer, and curves sweep the sparsity budget $L_0$ under the same steering and evaluation protocol. **Left:** Concept improvement, measuring how strongly the steered generations increase the target concept signal. **Middle:** Relative perplexity change, reporting the fluency/likelihood cost induced by steering relative to the unsteered baseline. **Right:** The combined steering score from the main text, which summarizes effectiveness and fluency into a single metric. This visualization matches the layout of Figure 9 and makes it easy to compare how steering trade-offs evolve with $L_0$ across layers and between the LLaMA-8B (autoregressive) and LLaDA-8B (diffusion) backbones.

Together, these plots make the layer-sparsity trade-offs of diffusion-time steering explicit: they show how increasing or decreasing $L_0$ shifts the balance between achieving the intended concept change and preserving fluent generation, and how this balance depends on intervention depth. As a result, the visualization serves as a practical guide for selecting layers and sparsity budgets when deploying SAE-based steering under a fixed evaluation protocol.

### F.2. Steering Ablations on Dream-7B Mask-SAEs

To illustrate how diffusion-time steering behavior changes under key hyperparameters, we present qualitative ablations on Dream-7B Mask-SAEs. The goal of this appendix is twofold: (i) to provide an intuitive, example-level complement to the quantitative steering metrics reported in the main text, and (ii) to help disentangle how steering strength, denoising duration, and per-step position selection jointly shape the final generation. Because Dream steering is applied repeatedly over the denoising trajectory (rather than once in a single left-to-right pass), these knobs can affect both the locality of the intervention and the accumulation of the intervention over time (how many times the feature direction is injected). We therefore vary three core controls: (i) the per-step position selector (`token_scope`), (ii) the steering strength (`amp_factor`), and (iii) the number of denoising steps (`dlm_steps`). All examples use neutral prefixes (Appendix E) to minimize prompt-specific bias; in each box we boldface words in the *after-steering* output that align with the feature explanation, making it easier to

visually attribute output changes to the intended feature direction rather than unrelated stylistic drift. For consistency with the main text, `amp_factor` corresponds to the steering coefficient $\alpha$ in Eq. (2).

*Table 4.* **Ablation settings.** Each row specifies: **amp_factor** (feature amplification strength; $\alpha$ in Eq. (2)), **dlm_steps** (number of denoising steps for DLM generation), **token_scope** (token-position selector; `all` or `top-K` positions per step for Dream), **n_prefix** (number of neutral prefixes per feature), **max_new_tokens** (generation length), and **Time/SAE** (time to evaluate one SAE under the setting).

| Model | amp_factor | dlm_steps | token_scope | n_prefix | max_new_tokens | Time/SAE |
|---|---|---|---|---|---|---|
| **Autoregressive baseline (no diffusion steps)** | | | | | | |
| Qwen2.5-7B | 2.0 | – | – | 5 | 30 | 13min |
| **Ablation: `token_scope`** (Dream-7B; `amp_factor=3.0`, `dlm_steps=30`) | | | | | | |
| Dream-7B | 3.0 | 30 | top-5 | 5 | 30 | 13min |
| Dream-7B | 3.0 | 30 | top-10 | 5 | 30 | 13min |
| Dream-7B | 3.0 | 30 | top-15 | 5 | 30 | 13min |
| **Ablation: `amp_factor`** (Dream-7B; `token_scope=all`, `dlm_steps=30`) | | | | | | |
| Dream-7B | 3.0 | 30 | all | 5 | 30 | 13min |
| Dream-7B | 2.0 | 30 | all | 5 | 30 | 13min |
| Dream-7B | 1.0 | 30 | all | 5 | 30 | 13min |
| **Ablation: `dlm_steps`** (Dream-7B; `amp_factor=3.0`, `token_scope=all`) | | | | | | |
| Dream-7B | 3.0 | 10 | all | 5 | 30 | 5min |
| Dream-7B | 3.0 | 30 | all | 5 | 30 | 13min |
| Dream-7B | 3.0 | 50 | all | 5 | 30 | 22min |

**`token_scope` (per-step position selector).**  For Dream-7B diffusion-time steering, `token_scope` controls *which token positions* receive the additive update at each denoising step. Using `top-K` restricts steering to the $K$ most-selected positions per step, while `all` applies steering to every position. Qualitatively, increasing $K$ tends to spread feature-consistent lexical effects more broadly, while smaller $K$ often concentrates the effect into a few salient positions.

---

**Example: varying `token_scope` (Dream-7B; `amp_factor=3.0`, `dlm_steps=30`)**

```
Feature:  activates on multi-channel audio / array terms (e.g., audio, sound, speakers,
channel, surround).
Prefix:  ``Once upon a time,''
without steering:  Once upon a time, in a galaxy far, far away, there was a tiny
planet ...
after steering (top-5):  Once upon a time, I tuned the audio just enough to make the
sound clear, then lowered the volume before the story continued.
after steering (top-10):  Once upon a time, the sound filled the room as the audio
played through two speakers, and every channel carried a slightly different echo.
after steering (top-15):  Once upon a time, we tested a surround audio setup where
each channel sent crisp sound to multiple speakers, and the audio stayed balanced even
when the volume rose.
```

---

**`amp_factor` (steering strength $\alpha$).**  `amp_factor` scales the magnitude of the feature-direction update and is exactly the coefficient $\alpha$ in Eq. (2). Larger values typically yield stronger insertion of feature-aligned tokens/phrases, but can also increase redundancy, off-topic drift, or exaggerated lexical markers of the feature.

---

**Example: varying `amp_factor` ($\alpha$) (Dream-7B; `token_scope=all`, `dlm_steps=30`)**

```
Feature:  activates on bounds / positional modifiers (e.g., upper, lower, bound,
limit).
without steering:  The answer depends on the context, and we can refine it with a
clearer definition and a concrete example.
after steering (amp_factor=1.0):  We can start with a simple constraint and set a soft
limit so the value stays reasonable.
after steering (amp_factor=2.0):  Next, we specify a lower bound to rule out trivial
cases, and also add an upper limit to keep the range finite.
after steering (amp_factor=3.0):  Finally, we enforce both a strict lower bound and a
strict upper limit, so the variable is tightly bounded within the permitted interval.
```

---

**`dlm_steps` (number of denoising steps).**  `dlm_steps` controls the diffusion compute budget during generation. With too few steps, we more often observe unstable reconstructions (e.g., format artifacts or rambling continuations), while moderate-to-large step counts produce steadier generations at the cost of higher latency.

---

**Example: varying `dlm_steps` (Dream-7B; `amp_factor`=3.0, `token_scope`=all)**

```
Feature:  activates on bounds / positional modifiers (e.g., upper, lower, bound,
limit).
Prefix:  ``For some reason,''
without steering:  For some reason, the results looked inconsistent, so we reran the
experiment and compared the outputs carefully.
after steering (dlm_steps=10):  For some reason, the report kept repeating constraints
like a lower threshold, an upper cutoff, another lower check, and yet another upper
limit in the same paragraph.
after steering (dlm_steps=30):  For some reason, the analysis mentioned a lower bound
and an upper limit once, then moved on to describe the rest of the reasoning in a more
balanced way.
after steering (dlm_steps=50):  For some reason, the write-up only briefly noted an
upper limit on the value before focusing on the main conclusion without emphasizing
bounds.
```

---

Overall, increasing `token_scope` (larger top-$K$ or `all`) distributes steering across more positions and makes feature-aligned terms appear more broadly in the output, while increasing `amp_factor` ($\alpha$) strengthens the feature effect and raises the frequency of feature-consistent tokens. In contrast, changing `dlm_steps` primarily affects generation stability: fewer steps tend to produce noisier outputs with more frequent (sometimes repetitive) feature markers, whereas more steps yield smoother text where the feature signal is typically weaker but more coherent.

## G. Decoding-Order Further Analysis Implementation

This appendix extends the decoding-order analysis in Section 5 with a complementary *Top-1 feature* diagnostic. While the main text focuses on Top-$K_{\text{feat}}$ set overlap and drift (Eqs. (14)–(15)), here we track whether the *single most dominant* SAE feature at each position remains stable across denoising steps, both *before decoding* (while the position is still masked) and *after decoding* (once the position has been filled).

**Top-1 feature identity.**  For decoding order $\mathcal{O}$, layer $\ell$, denoising step $k$, and token position $i$, let $h_{\ell,k,i}^{(\mathcal{O})} \in \mathbb{R}^k$ denote the SAE latent (Eq. (1)). We define the Top-1 feature identity as the largest-magnitude latent index:

$$f_{\ell,k,i}^{(\mathcal{O})} = \arg\max_{j\in[k]} \left| h_{\ell,k,i,j}^{(\mathcal{O})} \right|. \tag{16}$$

**Pre-decode Top-1 feature lock rate.**  Restricting to masked positions $\mathcal{M}(\mathbf{x}_{t_k}^{(\mathcal{O})})$ (Eq. (14)), we measure whether the Top-1 identity is unchanged between consecutive steps:

$$R_{\ell,k}^{\text{pre}}(\mathcal{O}) = \frac{1}{\left| \mathcal{M}(\mathbf{x}_{t_k}^{(\mathcal{O})}) \right|} \sum_{i\in\mathcal{M}(\mathbf{x}_{t_k}^{(\mathcal{O})})} \mathbf{1}\left[ f_{\ell,k,i}^{(\mathcal{O})} = f_{\ell,k-1,i}^{(\mathcal{O})} \right]. \tag{17}$$

**Post-decode Top-1 feature flip count.**  Let $\mathcal{U}(\mathbf{x}_{t_k}^{(\mathcal{O})}) = [N] \setminus \mathcal{M}(\mathbf{x}_{t_k}^{(\mathcal{O})})$ be decoded positions (Eq. (5)). We count how many decoded positions change their Top-1 identity between steps:

$$F_{\ell,k}^{\text{post}}(\mathcal{O}) = \sum_{i\in\mathcal{U}(\mathbf{x}_{t_k}^{(\mathcal{O})})} \mathbf{1}\left[ f_{\ell,k,i}^{(\mathcal{O})} \neq f_{\ell,k-1,i}^{(\mathcal{O})} \right]. \tag{18}$$

We compute $R_{\ell,k}^{\text{pre}}(\mathcal{O})$ and $F_{\ell,k}^{\text{post}}(\mathcal{O})$ on the same decoding-order runs as Section 5 and visualize them as layer-step heatmaps (each spanning the step range of its corresponding strategy).

Across ORIGIN, ENTROPY, and TOPK-MARGIN, the Top-1 feature identity is largely stable both before decoding (high lock rate on masked positions) and after decoding (few flips on decoded positions), indicating that the dominant semantic direction in SAE space is already highly consistent across denoising steps.

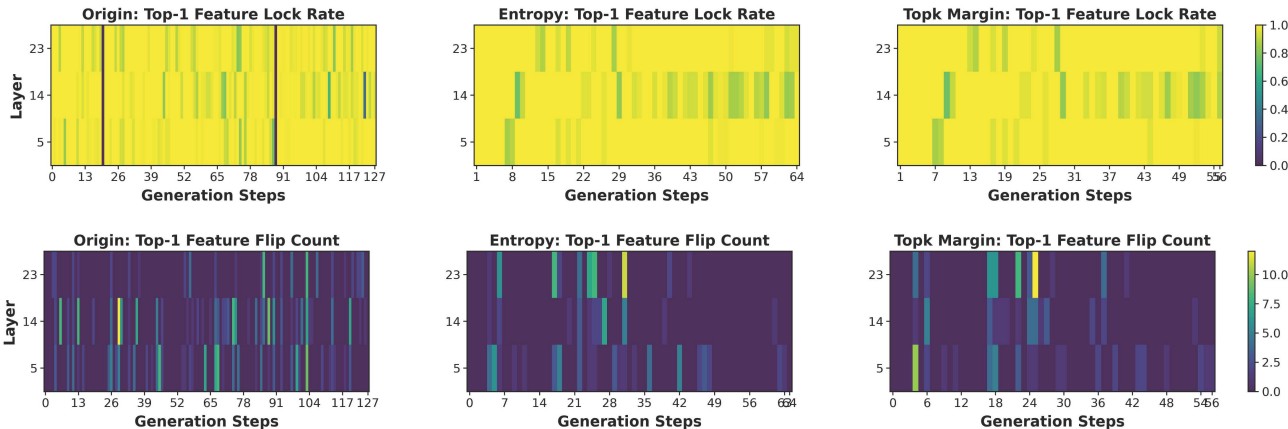

*Figure 11.* **Top-1 feature stability across decoding orders. Top:** pre-decode Top-1 lock rate $R_{\ell,k}^{\mathrm{pre}}(\mathcal{O})$ (Eq. 17), i.e., the fraction of masked positions whose Top-1 feature matches the previous step. **Bottom:** post-decode Top-1 flip count $F_{\ell,k}^{\mathrm{post}}(\mathcal{O})$ (Eq. 18), i.e., the number of decoded positions whose Top-1 feature changes between consecutive steps. Columns correspond to ORIGIN, ENTROPY, and TOPK-MARGIN; rows index tracked layers, and columns span denoising steps.

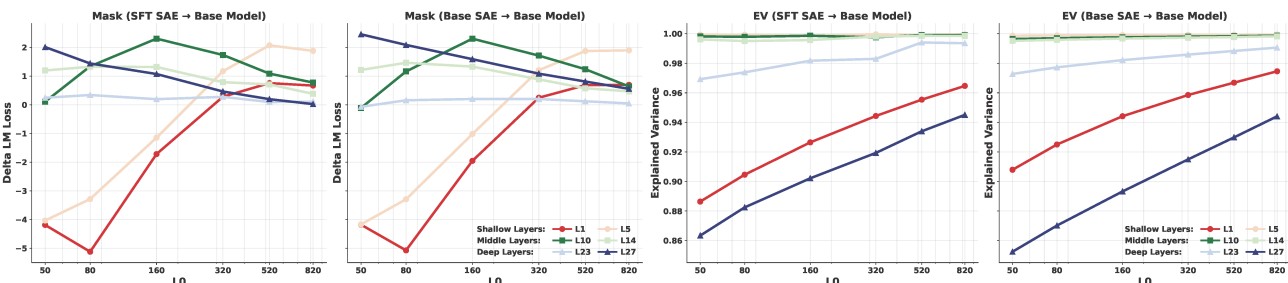

*Figure 12.* **Insertion of Base and SFT SAEs into the Dream Base backbone.** From left to right: (1) insertion-induced loss change $\Delta\mathcal{L}_{\mathrm{DLM}}$ when inserting SFT SAE into DREAM BASE; (2) $\Delta\mathcal{L}_{\mathrm{DLM}}$ when inserting BASE SAE into DREAM BASE; (3) explained variance (EV) for SFT SAE on DREAM BASE; (4) explained variance (EV) for BASE SAE on DREAM BASE. All panels sweep sparsity budget $L_0$ on the horizontal axis and plot per-layer curves (legend), with layers grouped into shallow, middle, and deep depths.

## H. SAE Insertion Results on Dream Base with Layerwise Explained Variance and $\Delta$LM Loss

Section 6 evaluates base-sft transfer by inserting SAEs into DREAM SFT. Here we provide the complementary setting: we insert the same pair of SAEs into the DREAM BASE backbone and report layerwise insertion metrics across a sweep of sparsity budgets $L_0$. Concretely, we compare a BASE SAE trained on DREAM BASE and an SFT SAE trained on DREAM SFT, while keeping SAE architecture and training protocol fixed. For each target layer, we evaluate reconstruction fidelity using explained variance (EV; Eq. (9)) and functional faithfulness using the delta loss change $\Delta\mathcal{L}_{\mathrm{DLM}}$ (Eq. (10)).

Figure 12 summarizes these insertion results on DREAM BASE across layers and $L_0$ settings. The left two panels report $\Delta\mathcal{L}_{\mathrm{DLM}}$ under masked-token denoising evaluation for inserting the SFT SAE and BASE SAE, respectively; the right two panels report the corresponding EV curves. Curves are grouped by depth (shallow/middle/deep) to highlight how insertion behavior changes across the network.

On DREAM BASE, inserting BASE SAE and SFT SAE yields very similar $\Delta\mathcal{L}_{\mathrm{DLM}}$ trends across layers and $L_0$, even though shallow-layer EV (notably L1) can differ more clearly between the two. This highlights that EV differences do not necessarily translate into proportional behavioral impact under insertion.

In the middle layers, both insertion behavior and EV remain closely matched over the $L_0$ sweep regardless of whether the SAE is trained on DREAM BASE or DREAM SFT, indicating strong cross-model transfer of mid-layer SAE subspaces.

