# OpenReview forum: "DLM-Scope: Mechanistic Interpretability of Diffusion Language Models via Sparse Autoencoders"
_ICML.cc/2026/Conference — ICML 2026 regular_

### Official Review · Reviewer_i8zb · 2026-02-17

**Soundness:** 2
**Presentation:** 4
**Significance:** 2
**Originality:** 1
**Overall Recommendation:** 4
**Confidence:** 4

**Summary:**

The authors adapt the popular Top-K SAEs from autoregressive language models to diffusion language models (DLMs). To handle the idiosyncrasies introduced by the diffusion-based generation, the authors introduce two possible strategies for steering the generation with SAEs, based on the presence of masks. For evaluation, the authors focus primarily on steering (adapted from AxBench), showing improved results over LLMs under their metric. Furthermore, qualitative evidence is provided in the form of standard automated interpretability techniques. Finally, the authors explore two applications of SAEs in the DLM setting, for analysis of decoding strategies and generalization ability to instruction-tuned models.

**Compliance With Llm Reviewing Policy:**

Affirmed.

**Final Justification:**

The authors have addressed all issues about missing evaluations and clarifications around framing of negative loss.

**Key Questions For Authors:**

## [q1] steering token choice differences

Around [L272] the authors use a different choice of all-tokens vs update-tokens for two different models, seemingly without explanation. Yet the authors state on [L195] that they "study" both. Might the authors expand on why this choice is made for one, but not the other? I believe a study of the newly introduced strategies should warrant reporting results for both strategies, for both models, so readers can understand the relevant strengths/weaknesses of each.

**Limitations:**

Limitations are not explicitly addressed.

The authors do a good job of describing the dual-use nature of steering in the impact statement. But some limitations of the methodology and experimental design are needed. Discussions of, for example, ways in which feature evaluations may be inadequate, or reliance on one LLM-as-judge might have its issues, etc, are some of the limitations i would like to see discussed.

**Strengths And Weaknesses:**

# Strengths
## [s1] Timely extension of interpretability techniques to DLMs

With the rise of DLMs, extending and stress-testing our current techniques for interpretability in the new language modeling paradigm is important work, and the paper offers a valuable contribution here.

## [s2] Thorough evaluation

Whilst i have a few reservations about specific evaluation protocols, the sheer number of experiments performed by the authors for the TopK-SAE is commendable. Specifically for Figure 3, the authors sweep over a large range of models, layers, and sparsity values. The authors provide exhaustive evidence of the sparsity-accuracy trade-off in a variety of settings that will surely be useful to many practitioners looking to do interpretability work in DLMs.

# Weaknesses

## [w1] Non-standard steering evaluation

The authors make the reasonable claim that they follow the steering evaluation of AxBench [L246]. However, rather than adopting AxBench's standard "fluency" and "instruct" scores, they introduce their own perplexity measure, in addition to the concept score. This deviation from the standard evaluation protocol appears to me, unaccounted for -- the authors should report the standard metric used in the literature so that readers can better place the results of the proposed method. Currently, strong claims are made around [L297] about their SAE steering improving over LLMs, but i do not think this is substantiated without the full AxBench metric.

Secondly, AxBench also reports steering scores for prompting. Given that the main proposed application/evaluation of the authors' DLM SAEs is steering, the authors should also report the standard steering score from prompting. I suspect that prompting performs much better than SAE steering in LDMs (as is the case in AxBench for LLMs), in which case a critical discussion of what benefit SAEs bring in DLMs would be necessary. Conversely, if SAE steering in DLMs improve over prompting under the standard AxBench metric, this is a significant result that should be highlighted by the authors -- either way a comparison is insightful.

## [w2] Autointerp scores alone are not sufficient

Recent work from Jan 2025 (now accepted at ICLR 2026) [heap-25] shows that automated interpretability metrics do not distinguish between trained and random SAE features. I believe it would strengthen the paper to adopt the full AxBench evaluation and include the standard quantitative concept detection to provide additional evidence, given the issues with autointerp.

## [w3] Negative loss framing

The authors note that occasionally the SAE approximation leads to *reduced* loss [L026,L184], which appears present only at shallow layers, in the high sparsity regime. I believe this is a concerning sign, but the authors currently pose the results almost as if this were a desirable property. If the goal of the SAEs is to learn a faithful approximation, shouldn't we care about the absolute value of any "difference" in loss?

## [w4] (minor) notation consistency

There are some small but important notational inconsistencies i believe need ironing out around what is a vector, and what is a scalar. [L092] introduce x presumably as vectors with regular font. This is fine if consistent, but then around [L151-L152], x is re-introduced with \mathbf{x} -- are these the same xs? The authors should clarify this and/or fix the inconsistencies throughout. My personal suggestion would be that the authors use the standard convention of bold font for vectors/matrices, and regular font for scalars. I think this is important, for example, when we look at Eq 2., where the operands being scalars or vectors changes entirely what computation is performed.

# References

* [heap-25]: Heap, Thomas, et al. "Automated Interpretability Metrics Do Not Distinguish Trained and Random Transformers." arXiv preprint arXiv:2501.17727 (2025).

---

> ### Author Rebuttal · Authors · 2026-03-31
>
> We sincerely appreciate the reviewer’s time and constructive comments!
> >W1: Non-standard steering evaluation
>
> In [L246], we explicitly state that we “follow the steering evaluation protocol [1][2].” Our metric design follows prior work by using ppl to assess steering fluency (Table 3 in [1]) and concept score to measure steering effectiveness, thus combining the strengths of both papers.
>
> Also, entirely following Table 2 of [2], we further report steering scores at middle and deep layers, and we additionally include AxBench-style prompt baseline for both DLM (Dream-7B) and LLM (Qwen2.5-7B):
> |Method|L10|L23|
> |-|-|-|
> |DLM-SAE|0.289|0.516|
> |LLM-SAE|0.221|0.219|
> |DLM-Prompt|1.078|1.061|
> |LLM-Prompt|1.175|1.196|
>
> These results further support our claim under AxBench: **SAE features enable diffusion-time interventions that are effective and often outperform LLM steering.** At the same time, the comparison also clarifies the boundary of the result: in DLMs, SAE steering still does not surpass the prompt baseline in absolute score, but **the gap is substantially smaller than in LLMs.**
>
> [1] LayerNavigator: Finding Promising Intervention Layers for Efficient Activation Steering in Large Language Models
>
> [2] AXBENCH: Steering LLMs? Even Simple Baselines Outperform Sparse Autoencoders
> >W2:  Autointerp scores alone are not sufficient
>
> Beyond Autointerp, we supplemented the full AxBench evaluation, including the Concept, Instruct, Fluency, and overall Steering scores.
> |Model|L10 Concept|L10 Instruct|L10 Fluency|L10 Steering|L23 Concept|L23 Instruct|L23 Fluency|L23 Steering|
> |-|-|-|-|-|-|-|-|-|
> |Qwen-2.5-7B|0.096|1.681|1.009|0.250|0.097|1.705|0.972|0.252|
> |Dream-Unmask|0.111|1.675|1.007|0.283|0.225|1.685|0.955|0.493|
> |Dream-Mask|0.117|1.654|0.978|**0.295**|0.258|1.647|0.923|**0.539**|
> |LLaMA-8B|0.105|1.406|0.855|0.263|0.123|1.454|0.785|0.297|
> |LLaDA-Unmask|0.134|1.533|0.923|0.326|0.236|1.450|0.856|**0.492**|
> |LLaDA-Mask|0.135|1.545|1.023|**0.332**|0.225|1.531|0.908|0.484|
>
> Follow the complete AxBench framework as suggested, the results in the table provide additional evidence and further support the conclusions of section 4 in this paper.
> >W3: Negative loss framing
>
> In our evaluation, we have two metrics: reconstruction fidelity, measured by explained variance (EV), and insertion-induced functional change, measured by ΔLM loss. The results from EV have demonstrated that DLM-SAE has reconstruction capabilities. Therefore, the ΔLM loss reveals a phenomenon unique to DLM, which is **enlightening for subsequent training of SAEs in DLM** rather than implying an ideal property.
>
> To further interpret the -ΔLM, we vary the mask ratio t in early Dream-7B Mask-SAE layers, testing whether low t resembles the LLM case while higher t reveals a diffusion-specific effect.
> |Mask Ratio t|DLM loss|ΔLM loss|
> |-|-|-|
> |0.1|6.526|0.506|
> |0.2|7.133|0.013|
> |0.3|7.513|-0.427|
> |0.4|7.658|-0.581|
> |0.5|7.631|-0.584|
>
> This ablation shows : when the mask ratio is small, ΔLM loss>0, which is similar to the usual LLM case and can be viewed as the near-minimal-mask limit; when the mask ratio is larger, ΔLM loss<0. This metric reveals a DLM-specific mechanistic phenomenon that becomes visible only in sufficiently corrupted denoising states.
>
>  **ΔLM loss offers a view for studying how DLM SAEs differ from LLM SAEs**. And more important thing is whether SAE yield interpretable, and controllable features. Our later sections (3.3, 4) show that they do.
> >W4: Notation consistency
>
> The $x$ at L102 and the $\mathbf{x}$ at L151–152 refer to the same object: the residual-stream activation. Eq. (2) will be rewritten with $\mathbf{x}$ and $\mathbf{v}_f$, with $\alpha$ and $m_f$ explicitly treated as scalars, and we will state these object types explicitly when first introduced.
> >Q1: Steering token choice differences
>
> The reason we use different strategies for two DLMs is tied to their architectures: Dream is an AR model adapted into diffusion, so it retains a stronger next-token fluency prior, whereas LLaDA is a from-scratch symmetric diffusion model without an inherent left-to-right bias. Consequently, ALL-TOKENS (A) steering is more likely to disrupt fluency in LLaDA, while Dream can better preserve it. Therefore, we introduce UPDATE-TOKENS (B) steering for LLaDA. We now report steering score of both strategies for both models under AxBench as suggested:
> |Method|L10|L23|
> |-|-|-|
> |Qwen2.5-7B|0.250|0.252|
> |Dream-SAE (A)|**0.289**|**0.516**|
> |Dream-SAE (B)|0.245|0.385|
> |LLaMA3-8B|0.263|0.297|
> |LLaDA-SAE (A)|0.201|0.275|
> |LLaDA-SAE (B)|**0.325**|**0.488**|
>
> We use the **architecture-motivated default** in the main table, which does not affect the core conclusion of Section 4. We will add these strategy comparisons for both models and clarify the motivation behind the design choice in the appendix.
> >L1: Limitations are not explicitly addressed.
>
> Following your suggestion, we will address the current feature evaluation protocol and the reliance on a single LLM judge.

---

> > ### Author Rebuttal · Reviewer_i8zb · 2026-04-01
> >
> > Thanks to the authors for their efforts!
> >
> > The new results on AxBench are a useful addition and indeed further support the authors' claim.
> >
> > > These results further support our claim under AxBench: SAE features enable diffusion-time interventions that are effective and often outperform LLM steering. At the same time, the comparison also clarifies the boundary of the result: in DLMs, SAE steering still does not surpass the prompt baseline in absolute score, but the gap is substantially smaller than in LLMs.
> >
> > This is a balanced and meaningful suggestion, and I believe the authors should include such a discussion of the pros and cons in the revised manuscript.
> >
> > > enlightening for subsequent training of SAEs in DLM rather than implying an ideal property.
> >
> > I also agree that this characterization of the loss reduction phenomenon is fair and reasonable! It would be useful to have this added to the paper when discussing the loss reduction throughout.
> >
> > The authors have addressed all my concerns, and I will update my score accordingly.

---

> > > ### Author Response · Authors · 2026-04-02
> > >
> > > Thank you for your positive assessment. We sincerely appreciate your careful review and constructive suggestions. In the revised version, we will incorporate all the points you raised through corresponding revisions and additions. We also confirm that all code will be fully open-sourced to ensure transparency and reproducibility.

---

### Official Review · Reviewer_o6fz · 2026-02-27

**Soundness:** 2
**Presentation:** 3
**Significance:** 3
**Originality:** 4
**Overall Recommendation:** 4
**Confidence:** 4

**Summary:**

This paper presents DLM-Scope, an SAE-based mechanistic interpretability framework for Diffusion Language Models. It shows that SAEs can faithfully reconstruct DLM activations, enable effective diffusion-time steering, explain decoding-order differences, and transfer across base and instruction-tuned models, with the exception of the deepest layers. Notably, early-layer SAE insertion can even reduce cross-entropy loss, an effect not observed in autoregressive LLMs.

**Compliance With Llm Reviewing Policy:**

Affirmed.

**Final Justification:**

The rebuttal provides meaningful additional evidence and partially addresses my main concerns, particularly through targeted ablations. While the mechanistic explanation remains incomplete and relies on proxy-level analysis, the empirical findings are compelling and better supported. I therefore update my score and lean toward a borderline accept.

**Key Questions For Authors:**

1. Mechanistic explanation of the negative ΔLoss phenomenon.
   The finding that SAE insertion in early DLM layers reduces cross-entropy loss is a central and novel divergence from autoregressive LLMs. Can the authors provide causal evidence clarifying whether this effect arises from denoising, implicit regularization, or alignment with the diffusion objective? For instance, have you conducted targeted ablations or analyzed activation statistics to isolate this mechanism?

2. iso-intervention steering comparison.
   DLM steering is applied across multiple denoising steps, while the LLM baseline uses a single-pass intervention. Can the authors provide a controlled comparison under matched intervention frequency or compute budget, such as a single-step DLM steering ablation? This is necessary to decouple intrinsic feature quality from cumulative multi-step refinement.

3. Full distribution of interpretability scores.
   The reported interpretability scores show notable variance, with some latents scoring as low as 0.31 (L23-615) and 0.286 (Feature 113). Can the authors provide the full score distribution across all latents and per-layer summary statistics? This would clarify whether low-scoring features are rare exceptions or indicate a systematic limitation in capturing DLM-specific representations.

**Limitations:**

Yes. The paper includes an Impact Statement and acknowledges potential misuse risks. The discussion could be further strengthened by explicitly elaborating on scalability limitations and potential side effects of multi-step steering, but the current treatment is generally adequate.

**Strengths And Weaknesses:**

## Strengths

- First SAE-based interpretability framework for DLMs.
Introduces diffusion-aware training strategies (e.g., MASK-SAE, UNMASK-SAE) that successfully extend mechanistic interpretability to discrete diffusion language models.

- Identification of a DLM-specific phenomenon.
Reveals that early-layer SAE insertion can reduce cross-entropy loss, a behavior not observed in autoregressive LLMs.

- Strong practical utility.
Demonstrates effective diffusion-time steering and provides mechanistic explanations for decoding-order performance differences.

## Weaknesses
- The finding that SAE insertion in early DLM layers reduces cross-entropy loss is interesting but insufficiently analyzed. The paper provides no causal evidence or targeted ablations to determine whether this effect arises from denoising, regularization, or alignment with the diffusion objective. Since this result constitutes a central divergence from autoregressive LLMs, the lack of mechanistic grounding substantially weakens the paper’s scientific depth.
- The steering comparison lacks computational alignment. DLM steering is applied across multiple denoising steps, while the LLM baseline uses a single-pass intervention. Without iso-intervention, it remains unclear whether the reported gains stem from better feature representations or from repeated injections. This calls into question the claim of inherent superiority in DLM-SAE steering.
- The automated interpretation scores show substantial variance, with some latents scoring below 0.3. Such low fidelity weakens the claim of consistent human-meaningful feature extraction. The lack of a full score distribution raises concerns about selection bias, and it remains unclear whether low scores stem from the describer or from intrinsic properties of DLM representations.

---

> ### Author Rebuttal · Authors · 2026-03-31
>
> We are grateful for the reviewer’s constructive feedback!
> >W1&Q1: Mechanism of -ΔLoss...provide evidence from denoising, regularization, or objective alignment?
>
> We run targeted ablations in the shallow Dream-7B MASK-SAE layers, where -ΔLM loss appears as you suggested. A full decomposition of “objective alignment” is difficult after SAE insertion (in DLMs, corruption level, masked-token recovery, and denoising representations are mixed together in the same forward pass). To show this, we use the closest causal proxies we could design:
>
>  (1) **Denoising: Variance Reduction Ratio,** i.e., how much the SAE reconstruction removes activation variance
>
>  (2) **Regularization: ΔLM loss (shrink/random).** Replace the SAE correction by a norm-matched shrink or random perturbation. If the gain is mainly regularization, ΔLM loss (shrink)≈ΔLM loss (SAE).
>
>  (3) **Objective alignment: ΔLM loss (full/parallel).** Keep only the component of the SAE correction parallel to the DLM-loss gradient. If the gain is mainly objective alignment, ΔLM loss (parallel)≈ΔLM loss (full).
> |Layer(L0)|Variance Reduction Ratio|ΔLM loss (SAE)|ΔLM loss (shrink)|ΔLM loss (random)|ΔLM loss (full)|ΔLM loss (parallel)|
> |-|-|-|-|-|-|-|
> |L1/L5(50)|0.232|-1.222|-0.309|-0.068|-1.076|0.051|
> |L1/L5(80)|0.201|-1.337|-0.229|-0.059|-1.199|0.040|
> |L1/L5(160)|0.167|-0.772|-0.169|-0.052|-0.788|0.046|
> |L1/L5(320)|0.134|-0.450|-0.088|-0.024|-0.370|0.022|
> |L1/L5(520)|0.118|-0.171|-0.063|-0.024|-0.139|0.017|
> |L1/L5(820)|0.104|-0.014|-0.055|-0.017|-0.015|0.016|
>
> **The ablations suggest that denoising partly explains -ΔLM loss, rather than generic regularization or simple gradient alignment.**  From table, shrink/random controls do not recover the gain, the parallel component stays near zero, and variance reduction tracks the effect.
>
> We further test whether this phenomenon is diffusion-specific by varying the mask ratio t in these layers:
> |Mask Ratio t|DLM loss|ΔLM loss|
> |-|-|-|
> |0.1|6.526|0.506|
> |0.2|7.133|0.013|
> |0.3|7.513|-0.427|
> |0.4|7.658|-0.581|
> |0.5|7.631|-0.584|
>
> When the mask ratio is small, ΔLM loss>0, similar to the usual LLM case; when the mask ratio is larger, ΔLM loss<0, revealing a DLM-specific effect. We conjecture **SAE directions are more helpful in DLMs when masked-token recovery becomes harder,** because they provide downstream layers with activations better suited for recovering missing tokens.
> >W2&Q2: Iso-intervention steering comparison...provide a controlled comparison under matched intervention frequency?
>
> Our claim is not that DLM SAEs learn better feature than LLM SAEs for steering. The key point is **DLM inference process is more naturally suited to steering**: conceptual signals can accumulate over multiple steps while preserving fluency. We also add single-step steering comparison for them. For DLM, we apply steering at only one denoising step (5/10/20 out of 30) and report steering score in the table with the AxBench protocol [1]:
> |DLM SAE|Step 5|Step 10|Step 20|AllSteps|LLM SAE|
> |-|-|-|-|-|-|
> |L1|0.046|0.038|0.058|0.044|0.092|
> |L5|0.079|0.079|0.095|0.134|0.137|
> |L10|0.100|0.093|0.106|0.289|0.221|
> |L14|0.121|0.090|0.113|0.248|0.203|
> |L23|0.187|0.192|0.159|0.516|0.219|
> |L27|0.311|**0.366**|0.305|0.319|0.114|
> |Average|0.146|0.153|0.146|**0.258**|0.164|
>
> The results show single-step DLM steering is slightly weaker than LLM steering on average, while some single-step settings can still perform strongly (0.366). We will revise this to avoid misunderstanding: **SAE features in DLMs enable effective steering, and DLM inference is more compatible with steering.**
>
> [1] AXBENCH: Steering LLMs? Even Simple Baselines Outperform Sparse Autoencoders
> >W3&Q3: Full distribution of interpretability scores...provide the full score distribution...per-layer summary statistics?
>
> We compute the full distribution of scores across all latents and add per-layer summary statistics for all DLM-SAE and LLM-SAE. Specifically, we report the mean, q25, median, q75 and the ratio of low-scoring features (< 0.28).
> |SAE|mean score|q25|median score|q75|<0.28 ratio (%)|
> |-|-|-|-|-|-|
> |DLM-SAE-ALL|0.733|0.571|0.857|0.929|1.0|
> |L1|0.798|0.714|0.857|1.000|0.8|
> |L5|0.775|0.714|0.857|0.929|1.0|
> |L10|0.764|0.714|0.857|0.929|0.5|
> |L14|0.676|0.357|0.786|0.857|0.5|
> |L23|0.701|0.357|0.857|0.929|2.0|
> |L27|0.682|0.321|0.786|0.929|1.0|
> |LLM-SAE-ALL|0.757|0.643|0.857|0.929|0.6|
> |L1|0.811|0.714|0.857|1.000|0.0|
> |L5|0.766|0.714|0.857|0.929|0.2|
> |L10|0.767|0.714|0.857|0.929|0.7|
> |L14|0.722|0.571|0.786|0.857|1.2|
> |L23|0.735|0.571|0.786|0.929|0.5|
> |L27|0.737|0.571|0.857|0.929|0.8|
>
> Overall, the score distributions are concentrated at relatively high values (0.7+), while low-scoring features are rare exceptions (typically around 0–2%), which supports that DLM-SAEs do extract interpretable features.
> >L1: Discussion could be further strengthened...
>
> We will further strengthen the discussion of explicitly elaborating on scalability limitations and potential side effects of multi-step steering.

---

> > ### Author Rebuttal · Reviewer_o6fz · 2026-04-03
> >
> > The rebuttal provides meaningful additional evidence and partially addresses my main concerns, particularly through targeted ablations. While the mechanistic explanation remains incomplete and relies on proxy-level analysis, the empirical findings are compelling and better supported. I therefore update my score and lean toward a borderline accept.

---

> > > ### Author Response · Authors · 2026-04-03
> > >
> > > Thank you for your thoughtful and encouraging feedback. We sincerely appreciate your recognition that the additional experiments and targeted ablations have strengthened the empirical support for our work. In the revised version, we will further investigate this phenomenon through more detailed experiments and analysis, and provide a clearer and more comprehensive discussion. We are grateful for your updated assessment and your consideration of our paper.

---

### Official Review · Reviewer_W1t8 · 2026-03-13

**Soundness:** 3
**Presentation:** 3
**Significance:** 3
**Originality:** 3
**Overall Recommendation:** 5
**Confidence:** 1

**Summary:**

This paper implements the dLLM version of the SAE to find interpretable directions. The resulting directions successfully steer the dLLM and shows generalizability for steering fine-tuned dLLMs.

**Compliance With Llm Reviewing Policy:**

Affirmed.

**Final Justification:**

My concerns are resolved. I will keep my current score.

**Key Questions For Authors:**

See my weakness sections.

**Limitations:**

Overall the method looks good and promising. The limitation I can think of is that training such a SAE for dLLM can be complex and difficult.

**Strengths And Weaknesses:**

Strengths:

This is a pioneering work that adapts SAE to dLLMs for mechanism interpretation. The authors explore a wide design space and conduct comprehensive experiments. The results look promising.

Weaknesses:

1. The SAE trained on base model does not work well when applied to the deepest layers. It is not clear why this happens.

2. Unlike SAE for LLM, training a SAE for all different timestep t seems to be a very complex and difficult task. Not sure if the authors encounter optimization difficulty.

---

> ### Author Rebuttal · Authors · 2026-03-31
>
> We are grateful for the reviewer’s constructive feedback, which helped us improve clarity!
>
> > W1: The SAE trained on base model does not work well when applied to the deepest layers. It is not clear why this happens.
>
> It's not that SAE is unavailable for the deepest layers, but rather that Base→SFT transfer only breaks at the single deepest layer L27.
>
> **(i) The deepest layers are already shown in the paper to be usable.** In our results, the deep layers  $L23/L27$ already satisfy the core criteria of a useful SAE interface: they remain functionally usable under insertion ($\Delta L_{DLM}$), reconstructive (high EV in Fig. 3), interpretable, and especially effective for steering, where the strongest gains appear at the deepest layer.
> |Criterion|Evidence|Takeaway|
> |-|-|-|
> |$\Delta L_{DLM}$|Fig. 3 reports a strong sparsity-fidelity trade-off for Dream-SAEs, with usable insertion behavior across shallow/middle/deep layers|Deep-layer insertion remains functionally meaningful|
> |EV|Fig. 3 shows high explained variance for deep layers $L23/L27$ across the $L0$ sweep|Deep-layer SAEs reconstruct residual states well|
> |Interpretability|Interp score is 0.705 in L23, 0.718 in L27, and approximately 0.72 in the shallow and intermediate layers.|Deep-layer features are still interpretable|
> |Steering|At $L0=80$, Dream-SAE reaches $S=0.20$ at $L23$ and **$S=0.34$** at $L27$, while the average steering score is $0.11$ in shallow layers and $0.09$ in middle layers.|Deep layers are not only usable, but strongest for intervention|
>
> **(ii) The deepest-layer result in Sec. 6 is therefore a very localized transfer exception, not evidence that the base-trained SAE fails in general.** The transfer experiment evaluates 6 layers $\{L1,L5,L10,L14,L23,L27\}$, only **$L27$** shows a clear separation in $\Delta L_{DLM}$.
> |Transfer check|Observation|
> |-|-|
> |Layers tested|$L1,L5,L10,L14,L23,L27$|
> |Main pattern|BASE SAE and SFT SAE are nearly identical on $L1$–$L23$|
> |Exception|Clear separation appears only at $L27$|
> |Rollout stress test|30 denoising steps, 30 generated tokens|
> |Rollout conclusion|Reuse holds for shallow/middle layers; mismatch is concentrated at $L27$|
>
> **We interpret this result as a top-layer transfer boundary induced by instruction tuning/alignment sensitivity, not as evidence that SAEs fail in the deepest layers.** Because the mismatch is localized to the single deepest layer $L27$ and becomes more pronounced under instruction rollouts, while $L1$–$L23$ remain nearly unchanged between BASE SAE and SFT SAE.
>
> What breaks is specifically Base→SFT transfer, which holds for 5 of 6 tested layers and fails only at the single deepest layer, indicating that post-training most strongly reshapes the deepest-layer subspace and introduces instruction-critical directions not captured by the base SAE. Therefore, this phenomenon is itself a mechanistically informative finding: in our setting, post-training changes are concentrated most strongly in the deepest-layer.
> > W2: Training a SAE for all different timestep t seems to be a very complex and difficult task. Not sure if the authors encounter optimization difficulty.
>
> **(i) Our DLM-SAE training does not require a separate SAE for each diffusion timestep; it trains one shared SAE per layer under the DLM denoising distribution.**
>
> Relative to LLM-SAEs, the key difference is in the training distribution and objective. Standard LLM-SAE training uses activations from fully observed prefixes under causal masking, whereas DLM-SAE training must follow the DLM denoising objective in Eq. (3): each forward pass is conditioned on a random corruption level $t$ and a partially masked input $x_t$.
>
> Compared with LLM-SAEs, the added complexity is not timestep-wise optimization, but learning from timestep-mixed activations and explicitly deciding which positions should provide training signals under corruption.
>
> **(ii)** Below are two DLM-specific difficulties and corresponding design choices:
>
> |Difficulty|Our solution|
> |-|-|
> |Activations come from many corruption levels $t$|Train one shared SAE per layer under Eq. (6), rather than one SAE per timestep|
> |Corruption changes which token positions are meaningful|Introduce two controlled variants: MASK-SAE and UNMASK-SAE, as defined through Eq. (5) and Eq. (6)|
>
> **Empirically, this training recipe is stable and reproducible rather than optimization-fragile.** We successfully train 36 SAEs per backbone (6 layers × 6 sparsity settings) on both Dream-7B and LLaDA-8B, and observe consistent sparsity–fidelity trade-offs across models and layers.
>
> **(iii)** We also summarize the resource below.
>
> |Model|Disk / SAE|Training time / SAE|Peak memory / SAE|
> |-|-|-|-|
> |Qwen2.5-7B|449M|4h-9h|20G-27G|
> |Dream-7B|449M|3h-8h|20G-27G|
> |LLaMA3-8B|512M|4h-15h|20G-27G|
> |LLaDA-8B|512M|3h-13h|20G-27G|
>
> Our experiments show that this design yields a stable, reproducible TopK-SAE pipeline in practice, and **no additional computational cost or training time is introduced.**

---

> > ### Author Rebuttal · Reviewer_W1t8 · 2026-04-02
> >
> > My concerns are resolved. I will keep my current score.

---

> > > ### Author Response · Authors · 2026-04-02
> > >
> > > We sincerely appreciate your thoughtful review and constructive suggestions, which have helped improve the clarity of our work. In the revised version, we will incorporate all the points you raised and provide additional explanations and supporting details, including an expanded appendix with clearer and more accessible guidance for researchers who wish to train, study, and use this tool. We also commit to fully open-sourcing all code to support transparency and reproducibility.

---

### Official Review · Reviewer_1GqX · 2026-03-24

**Soundness:** 3
**Presentation:** 2
**Significance:** 3
**Originality:** 3
**Overall Recommendation:** 5
**Confidence:** 3

**Summary:**

The paper presents DLM-Scope, a novel framework that integrates Sparse Autoencoders (SAEs) with Diffusion Language Models (DLMs) as an interpretability tool for DLMs. To the best of our knowledge, this is the first work to combine SAEs with DLMs. DLM-Scope leverages both Mask-SAE and Unmask-SAE to capture different types of features. In contrast to standard large language models, splicing SAEs into the shallow layers of a DLM can reduce cross-entropy loss. Moreover, DLM-Scope allows targeted SAE features to be inserted at multiple denoising steps, demonstrating strong effectiveness for diffusion-time steering. The paper also investigates the impact of different decoding orders on SAE features. Finally, the results show that an SAE trained on a base DLM can be transferred to an instruction-tuned variant of the same model.

**Compliance With Llm Reviewing Policy:**

Affirmed.

**Final Justification:**

The research question is first work to apply Sparse Autoencoders to Diffusion Language Models in order to improve the interpretability of DLMs. Although I commented on several weaknesses, the authors have incorporated more experimental results and explanations into their paper to address the comments.

**Key Questions For Authors:**

- How does the model performance change when varying the number of sparse features $k$ in SAE? Would a larger $k$ yield more interpretable DLM features? Is there an optimal choice of $k$ for the proposed model?

- Could the authors provide more quantitative metrics to evaluate the interpretability of the extracted features instead of only showing qualitative examples in Appendix D?

- Could the authors give more instructions on choosing DLM decoding order: ORIGIN, TOPK-MARGIN, or ENTROPY? Aside from this, could the authors offer more theoretical insights into the more organized progression of confidence-based orders (TOPK-MARGIN and ENTROPY)?

- Please define $m_f$ in Eq. 2 and Eq. 7.

**Limitations:**

The authors have included societal impact, but do not adequately discuss the methodological limitations of their work.

**Strengths And Weaknesses:**

## Strengths
- The paper is clearly written and well organized, making it easy to follow.
- The research question is novel. To the best of the authors’ knowledge, this is the first work to apply Sparse Autoencoders to Diffusion Language Models in order to improve the interpretability of DLMs.

## Weaknesses
- It would improve readability if the authors provided more intuitive explanations before Equation (3) and Equation (6).
- The claim that “DLM steering often outperforms LLM steering” appears overstated. While the results support the effectiveness of DLM steering, the comparison with LLM steering is not entirely fair, since DLM steering intervenes across multiple denoising steps, whereas LLM steering typically intervenes at only a single step.
- The experiments are limited to Top-K SAEs and discrete, mask-based diffusion models. To better demonstrate the robustness and generalizability of the proposed framework, the authors should evaluate additional SAE architectures [1] and other diffusion language models [2].
- The current experimental results do not sufficiently support the claim that the SAE extracts interpretable features. To better validate this conclusion, the authors could report the distribution of auto-interpretation scores or the average score across all features.
- The claim of a “2–10× improvement” in line 301 on page 6 does not appear to be supported by Table 2.
- Section 4 would benefit from additional explanation and discussion.

[1] Rajamanoharan, Senthooran, et al. “Improving Sparse Decomposition of Language Model Activations with Gated Sparse Autoencoders.” Advances in Neural Information Processing Systems 37 (2024): 775–818.

[2] Li, Xiang, et al. “Diffusion-LM Improves Controllable Text Generation.” Advances in Neural Information Processing Systems 35 (2022): 4328–4343.

---

> ### Author Rebuttal · Authors · 2026-03-31
>
> We greatly appreciate reviewer’s helpful comments!
> >W1: Provide explanations before Equation (3) and (6)
>
> We will clarify Eq. (3) as masked-token prediction and Eq. (6) as reconstructing activations from masked/unmasked positions with SAE.
> >W2: Comparison with steering is not fair, since DLM steering...multiple denoising steps, whereas LLM steering...a single step
>
> Our point is **DLM inference is better suited to steering**, as control signals accumulate over steps. We also add a single-step comparison. For DLM, we apply steering at only one denoising step (5/10/20 out of 30) and report **steering score** with AxBench protocol [1]:
> |DLM|5|10|20|AllSteps|LLM|
> |-|-|-|-|-|-|
> |L1|0.046|0.038|0.058|0.044|0.092|
> |L5|0.079|0.079|0.095|0.134|0.137|
> |L10|0.100|0.093|0.106|0.289|0.221|
> |L14|0.121|0.090|0.113|0.248|0.203|
> |L23|0.187|0.192|0.159|0.516|0.219|
> |L27|0.311|**0.366**|0.305|0.319|0.114|
> |Average|0.146|0.153|0.146|**0.258**|0.164|
>
> Single-step DLM steering is slightly weaker on average, and some settings are even strong (0.366). We will clarify **DLM-SAE features enable effective steering, and DLM inference is more compatible with it.**
>
> [1] AXBENCH: Steering LLMs? Even Simple Baselines Outperform Sparse Autoencoders
> >W3: Experiments are limited to Top-K SAEs...should evaluate additional SAE architectures...
>
> We add two more SAE architectures, Gated and JumpReLU, on Dream-7B across Shallow (L1, L5), Middle (L10, L14), and Deep (L23, L27) layers under six sparsity levels (72 SAEs), using the same Section 3 pipeline:
> |Arch|ΔLM loss_S|ΔLM loss_M|ΔLM loss_D|EV_S|EV_M|EV_D|Interp_S|Interp_M|Interp_D|
> |-|-|-|-|-|-|-|-|-|-|
> |Gated|-3.121|0.414|0.366|0.968|0.994|0.985|0.752|0.707|0.784|
> |JumpReLU|-1.737|-0.913|0.306|0.964|0.998|0.962|0.730|0.703|0.775|
>
> Results are **consistent** with our claims: both architectures reproduce -ΔLM loss effect, achieve strong EV, and yield reasonable interpretability. We have fully validated framework on two discrete DLMs. Although we have not yet run pipeline on continuous DLMs (due to time constraints), but our method is general and provides a foundation for that setting.
> >W4&Q2: Report distribution of auto-interp scores...Provide quantitative metrics to evaluate interp...
>
> We now report score distributions and per-layer statistics for DLM-SAEs and LLM-SAEs: mean, q25, median, q75, and low-score fraction (<0.28).
> |SAE|mean|q25|median|q75|<0.28 ratio (%)|
> |-|-|-|-|-|-|
> |DLM-SAE-ALL|0.733|0.571|0.857|0.929|1.0|
> |L1|0.798|0.714|0.857|1.000|0.8|
> |L5|0.775|0.714|0.857|0.929|1.0|
> |L10|0.764|0.714|0.857|0.929|0.5|
> |L14|0.676|0.357|0.786|0.857|0.5|
> |L23|0.701|0.357|0.857|0.929|2.0|
> |L27|0.682|0.321|0.786|0.929|1.0|
> |LLM-SAE-ALL|0.757|0.643|0.857|0.929|0.6|
> |L1|0.811|0.714|0.857|1.000|0.0|
> |L5|0.766|0.714|0.857|0.929|0.2|
> |L10|0.767|0.714|0.857|0.929|0.7|
> |L14|0.722|0.571|0.786|0.857|1.2|
> |L23|0.735|0.571|0.786|0.929|0.5|
> |L27|0.737|0.571|0.857|0.929|0.8|
>
> Overall, scores are high (0.7+), with few low-scoring features (typically 0–2%), supporting DLM-SAE interpretability.
> >W5&W6: “2–10× improvement” does not...supported by Table 2. Section 4...benefit from additional explanation and discussion.
>
> Our claim was intended for **deep layers** in [L301], where Table 2 and AxBench results (Refer to W2's reply above) support 2× gains at L23/L27. We will add additional explanation and discussion for Section 4 in appendix.
> >Q1: How does model performance change when varying the number of k in SAE?
>
> We vary SAE scale on Dream-7B at 6 layers and report averaged results for Shallow (1,5), Middle (10,14), and Deep (23,27) layers. We test whether larger SAEs improve reconstruction, faithfulness, or interpretability.
> |Scale|ΔLM loss_S|ΔLM loss_M|ΔLM loss_D|EV_S|EV_M|EV_D|Interp_S|Interp_M|Interp_D|
> |-|-|-|-|-|-|-|-|-|-|
> |16K|-0.377|1.129|0.742|0.973|0.997|0.942|0.787|0.720|0.692|
> |32K|-0.584|1.086|0.742|0.975|0.997|0.940|0.716|0.715|0.717|
> |64K|-0.551|1.116|0.708|0.975|0.997|0.942|0.716|0.716|0.717|
>
> As scale increases, shallow-layer ΔLM loss becomes slightly more negative, while EV is nearly unchanged and interpretability is more stable across depth. Overall, 16K already provides a strong quality-efficiency trade-off.
> >Q3: Give instructions on choosing DLM decoding order...offer theoretical insights...
>
> A universal selection rule is beyond our scope. As in [2], ORIGIN suits simpler settings, while confidence-based orders are generally better for harder tasks.
>
> As suggested in [2], this behavior may stem from how confidence-based schedules interact with data distribution. Analyzing this mechanism is beyond our scope. Instead, we show SAE-based dynamics correlate with task performance across decoding orders.
>
> [2] Dream 7B: Diffusion Large Language Models
> >Q4&L1: Define m_f in Eq. 2 and Eq. 7...discuss methodological limitations
>
> We will define $m_f$ as per-sample scaling factor controlling intervention magnitude, and clarify methodological limitations such as imperfect automatic proxies and limited settings.

---

> > ### Author Rebuttal · Reviewer_1GqX · 2026-04-03
> >
> > My concerns have been adequately addressed. I will adjust my score accordingly.

---

### Decision · Program_Chairs · 2026-04-30

**Decision:**

Accept (regular)

**Comment:**

This paper proposes DLM-Scope, a framework for applying sparse autoencoders (SAEs) to diffusion language models (DLMs) in order to enable mechanistic interpretability and model steering. The key idea is to adapt SAE-based feature extraction to the diffusion setting, where representations evolve across denoising steps. The paper reports that SAE features can reconstruct activations, provide interpretable directions, and enable controllable generation, while also revealing diffusion-specific behaviors (e.g., differences across layers and timesteps).

Strengths:
(1) Reviewers agree that applying SAEs to DLMs is a relatively new direction and extends mechanistic interpretability to an emerging model class.
(2) Comprehensive empirical study. The paper includes: experiments across multiple layers, timesteps, and models, evaluation of reconstruction, interpretability, and steering, analysis of diffusion-specific behaviors.
(3) The method demonstrates controllable generation via SAE-based interventions, and insights into internal structure (e.g., layer-wise differences, transfer effects).

Reviewers also identified some weakness, including limited exploration of hyperparameters and settings, unclear generalization across architectures, and some instability in results across layers and experiments. Also, several findings (e.g., reduction in cross-entropy loss, behavior across layers) are empirically observed but not well explained. The mechanism behind these effects remains unclear.